# Glocal Information Bottleneck for Time Series Imputation

**Jie Yang**[1,2,*], **Kexin Zhang**[2,*], **Guibin Zhang**[3], **Philip S. Yu**[1], **Kaize Ding**[2,†]

[1]University of Illinois Chicago
[2]Northwestern University
[3]National University of Singapore
✉ Primary contact: `jyang265@uic.edu`

## Abstract

Time Series Imputation (TSI), which aims to recover missing values in temporal data, remains a fundamental challenge due to the complex and often high-rate missingness in real-world scenarios. Existing models typically optimize the point-wise reconstruction loss, focusing on recovering numerical values (local information). However, we observe that under high missing rates, these models still perform well in the training phase yet produce poor imputations and distorted latent representation distributions (global information) in the inference phase. This reveals a critical optimization dilemma: current objectives lack global guidance, leading models to overfit local noise and fail to capture global information of the data. To address this issue, we propose a new training paradigm, **Glocal I**nformation **B**ottleneck (**Glocal-IB**). Glocal-IB is model-agnostic and extends the standard IB framework by introducing a Global Alignment loss, derived from a tractable mutual information approximation. This loss aligns the latent representations of masked inputs with those of their originally observed counterparts. It helps the model retain global structure and local details while suppressing noise caused by missing values, giving rise to better generalization under high missingness. Extensive experiments on nine datasets confirm that Glocal-IB leads to consistently improved performance and aligned latent representations under missingness. Our code implementation is available in `https://github.com/Muyiiiii/NeurIPS-25-Glocal-IB`.

## 1 Introduction

Missing values are pervasive in real-world time series due to device malfunctions, transmission failures, and manual collection errors [52, 72]. These missing values occur with varying rates and patterns across domains such as healthcare [69, 41, 43], transportation [25], and energy systems [18, 20], thus substantially impairing the integrity of time series data and the performance of downstream tasks [17, 65]. Consequently, Time Series Imputation (TSI), which aims to reconstruct missing values from partially observed data, has emerged as a critical problem with broad practical significance [51].

Missing values disrupt the original structure of time series data, acting as structured noise that corrupts temporal dependencies and statistical patterns [27, 34]. To address this, existing TSI methods typically adopt encoder-decoder architectures [30, 6], trained by randomly masking observed values to simulate missingness [58, 13, 42]. The goal is to learn the global data distribution from corrupted observations, enabling the model to reconstruct masked values during training and serve as a conditional generative model for imputation at inference time [56, 53]. However, a critical optimization dilemma has emerged in this paradigm: Under high missing rates, models achieve training losses comparable

---

[*]Work done during internship at Northwestern University.
[†]Corresponding author.

39th Conference on Neural Information Processing Systems (NeurIPS 2025).

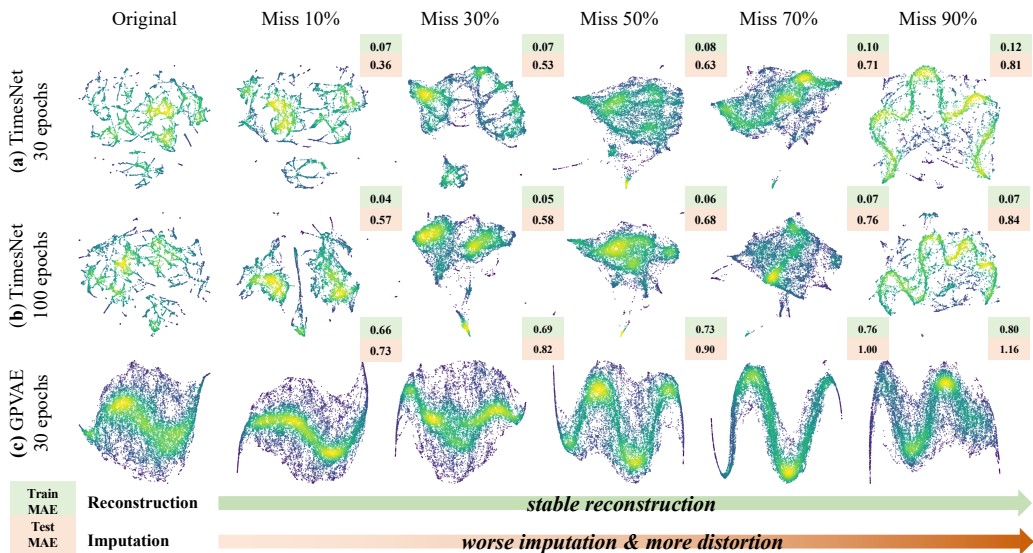

Figure 1: **Illustration of optimization dilemma in TSI.** We visualize the latent space of two representative models—TimesNet (a-b) and GPVAE (c)—trained under different missing rates and training epochs. Training and test losses are shown in green and orange boxes, respectively.

to low-missingness scenarios, suggesting successful convergence, yet suffer drastic performance degradation in imputation quality. To investigate this discrepancy, we conduct an empirical analysis of representative TSI models under varying missing rates. Our results highlight a gap between latent representations learned during training and their utility for accurate imputation. Fig. 1 (a-b) illustrates this phenomenon using TimesNet [60] (a-b) and GPVAE [14] (c) on the ETTh1 dataset [75]; additional results are provided in the Appendix D.1. Our findings highlight two key phenomena:

❶ *Low training loss does not necessarily imply good imputation.* As the missing rate increases, the model still performs well in reconstructing the training data, but their inference-time imputation quality drops substantially. Even more surprisingly, reducing the number of training epochs (resulting in slightly higher training loss) achieves better imputation results during inference. This suggests that the training objective under high missingness fails to guide the model toward generalizable representations, encouraging memorization of local observations rather than learning meaningful global structures and information.

❷ *Well-aligned representations are strongly related to good imputation.* We further visualize the latent space distributions of the models and observe that better imputation performance corresponds to representations that remain well-aligned with those derived from fully observed data. However, as missingness increases, the distributions become increasingly distorted, despite low training losses. This distortion correlates with poor imputation, suggesting that reconstruction losses (e.g., MAE/MSE) fail to preserve globally coherent structure under severe missingness.

These observations suggest that a fundamental limitation of current TSI methods lies in their training objectives. The focus on local numerical accuracy at each timestamp makes these models sensitive to temporal noise and redundant patterns [53, 7, 74], hindering their ability to capture the underlying global distribution. To address this issue, researchers have explored the Information Bottleneck (IB) principle [19, 1, 47], which encourages representations that discard irrelevant noise while preserving task-relevant information. However, most IB-based TSI methods [14, 7] still rely on local reconstruction loss to increase task-relevant mutual information,which is inadequate for capturing global structure. As a result, these models remain vulnerable to the same optimization dilemma. For example, GPVAE [14], as shown in Fig. 1 (c), suffers from severe latent space distortion and performance degradation as the missing rate increases. Its MAE degrades to 0.8, similar to the non-IB-based TimesNet [60] under the same conditions. Therefore, a key research question is raised: *Can we design a training paradigm that encourages TSI models to capture both global and local information from incomplete data, without overfitting to noise?*

**Our approach.** To answer this question, we propose a new training paradigm, **Glocal I**nformation **B**ottleneck (**Glocal-IB**), which is based on the trade-off between compactness (suppressing noise) and informativeness (preserving both global and local information). Unlike previous IB-based methods [35, 21, 32] that rely solely on reconstruction losses to increase the mutual information between latent representations and imputation targets, Glocal-IB goes one step further. Specifically, it extends the standard IB framework by introducing a Global Alignment loss, derived from a tractable mutual information approximation. This loss aligns the latent representations of masked inputs and their corresponding original inputs. Remarkably, Glocal-IB requires only a single Multilayer Perceptron (MLP) to implement the alignment loss, making it model-agnostic and easily integrable into existing encoder-decoder frameworks.

The main contributions of this paper are as follows:

- We identify a critical optimization dilemma in existing TSI methods: under high missing rates, models achieve low training loss but fail to learn globally semantic latent representations, leading to substantial degradation in imputation quality and severe latent space distortion.

- We propose a novel IB-based training paradigm, Glocal-IB, which explicitly enforces latent space consistency via a lightweight global alignment loss, alongside local reconstruction, thereby improving both global and local feature learning while removing irrelevant noise.

- Our empirical results validate the effectiveness of Glocal-IB. On nine benchmark datasets, it consistently achieves top imputation performance and helps form a smooth, structured latent space when applied to a vanilla Transformer. Similar improvements are observed across other backbones, showing strong generalization and better robustness under high missing rates.

## 2 Related Work

**Time Series Imputation** TSI has received increasing attention due to its critical impact in real-world applications [44, 76, 61]. Most recent methods adopt an encoder-decoder architecture [53], differing mainly in how they capture temporal dependencies. RNN-based models like GRU-D [4] and BRITS [3] handle temporal decay and bidirectional inference, while transformer-based models such as ImputeFormer [36] use attention mechanisms for long-range temporal modeling. Spatial correlations across variables are modeled by methods like GRIN [8] and SPIN [31], which integrate Graph Neural Networks (GNNs) [24, 49]. Moreover, CSDI [46], GPVAE [14], CIB [7], and USGAN [33] are proposed to learn probability distributions from the observed data. Despite architectural progress, most models remain sensitive to noise and temporal redundancy in the observed values [42, 57], due to the point-wise reconstruction loss. To address this, we introduce a new training paradigm, Glocal-IB, with a dual emphasis on global structure and local detail, thereby encouraging semantically stable representations and improving imputation under severe missingness.

**Optimization Dilemma** Recent studies have observed a mismatch between low training loss and poor test-time performance. For instance, in latent diffusion models (LDMs) for computer vision [12, 67, 70], high-capacity models produce over-concentrated latent spaces that capture low-level details at the cost of semantic coherence. Solutions to this in computer vision, such as VA-VAE [67] and REPA [70], leverage vision foundation models [39, 15, 16] to align the latent space, promoting richer semantics. However, time-series foundation models [45, 29, 54] are primarily trained with predictive or reconstruction losses and lack sufficient semantic information needed to mitigate this issue. To address this, we introduce a Global Alignment objective that encourages the latent representations of masked observed sequences to remain close to those of their original observed counterparts. In addition, compared to solutions that rely on foundation models, our approach is lightweight and efficient, requiring only one extra MLP.

## 3 Methodology

**Problem Definition.** Given an original multivariate time series $X = \{x_{1:T}^i \mid i = 1, \ldots, N\} \in \mathbb{R}^{N \times T}$, where $N$ is the number of variables and $T$ is the sequence length. To simulate missingness, a binary mask $M \in \{0, 1\}^{N \times T}$ is applied, where $M^{i,t} = 1$ indicates that $x_t^i$ is observed and $M^{i,t} = 0$ indicates it is missing. The masked input is then defined as $X^o = X \odot M$. TSI models outputs the imputed result $\hat{X} \in \mathbb{R}^{N \times T}$ based on the $X^o$, aiming at estimating the missing values in $X^o$.

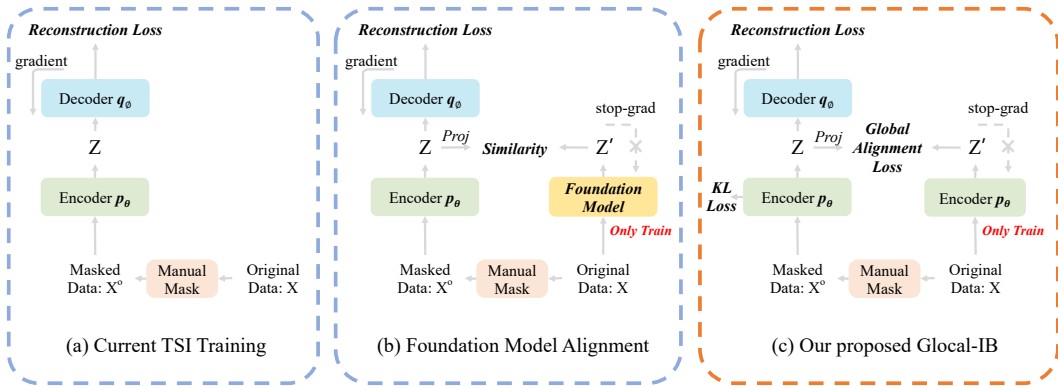

Figure 2: Framework comparison of three TSI training paradigms. Three paradigms differ in how to deal with the latent representations and how the key encoder and decoder are updated. (**a**): The encoder and decoder are updated end-to-end by back-propagation of reconstruction loss. (**b**): The latent representations are aligned with a frozen time series foundation model with original data. (**c**): *Glocal-IB* utilizes the encoder itself and a KL divergence to regularize the latent representations.

Throughout this paper, we refer to $X$ as the imputation target, $X^{\mathrm{o}}$ as the masked input, and $\hat{X}$ as the model's imputation result.

**Information Bottleneck Theory for Time Series Imputation** IB principle provides a theoretical framework for identifying informative parts of the input by balancing two competing objectives: compactness (regularization) and informativeness (task performance). This trade-off shapes the latent representations to retain only the essential structure for solving a specific task.

Let $X^{\mathrm{IB}}$ and $Y^{\mathrm{IB}}$ denote the original input data and the targets of a specific task, respectively. To get the balance between regularizing input data and maintaining good performance, there is a well-designed formula about $X^{\mathrm{IB}}$, $Y^{\mathrm{IB}}$, and the bottleneck variable $Z^{\mathrm{IB}}$ as follows:

$$\min [I(Z^{\mathrm{IB}}; X^{\mathrm{IB}}) - \beta \cdot I(Y^{\mathrm{IB}}; Z^{\mathrm{IB}})], \tag{1}$$

where $I(Z^{\mathrm{IB}}; X^{\mathrm{IB}})$ and $I(Y^{\mathrm{IB}}; Z^{\mathrm{IB}})$ represent the mutual information of $(Z^{\mathrm{IB}}, X^{\mathrm{IB}})$ and $(Y^{\mathrm{IB}}, Z^{\mathrm{IB}})$, and $\beta \in \mathbb{R}$ is a a Lagrange multiplier that balance the two mutual information. This offers a good understanding of what contributes most to the task from an information-theoretic perspective. Furthermore, according to the previous IB literature [1, 64], we can assume a factorization of the joint distribution as follows:

$$p(X^{\mathrm{IB}}, Y^{\mathrm{IB}}, Z^{\mathrm{IB}}) = p(Z^{\mathrm{IB}}|X^{\mathrm{IB}}, Y^{\mathrm{IB}})p(Y^{\mathrm{IB}}|X^{\mathrm{IB}})p(X^{\mathrm{IB}}) = p(Z^{\mathrm{IB}}|X^{\mathrm{IB}})p(Y^{\mathrm{IB}}|X^{\mathrm{IB}})p(X^{\mathrm{IB}}), \tag{2}$$

namely, there is a Markov chain $Y^{\mathrm{IB}} \leftrightarrow X^{\mathrm{IB}} \leftrightarrow Z^{\mathrm{IB}}$, indicating that the latent representations $Z^{\mathrm{IB}}$ can not directly depend on the targets $Y^{\mathrm{IB}}$. Then, following Eq. (1), we can define the TSI as a supervised IB task as follows:

$$\min_{\theta, \phi}[I_\theta(Z; X^{\mathrm{o}}) - \beta \cdot I_\phi(X; Z)], \tag{3}$$

where $\beta \in \mathbb{R}$ and $Z \in \mathbb{R}^{N \times d_{\mathrm{model}}}$ denote a preset hyperparameter and the latent representations, respectively. $\theta$ and $\phi$ denote the learnable parameters of the encoder $p_\theta(\cdot)$ and the decoder $q_\phi(\cdot)$ of our method Glocal-IB. Therefore, we can accomplish the TSI tasks by modeling crucial information from the partially observed data while filtering out redundant noise.

## 3.1 Overview

In this section, we introduce our proposed training paradigm **Glocal-IB**, which is grounded in the IB principle, and present the derivation of two components in Eq. 3. As shown in Fig. 2 (c), Glocal-IB is simple to use, adds only one MLP projector for alignment, and can be applied to a wide range of existing methods. Glocal-IB aims to balance two goals in the latent space: reducing noise and retaining both global and local information. To achieve this, it minimizes the mutual information between the masked input $X^{\mathrm{o}}$ and the latent representations $Z$, which helps remove noise introduced by incomplete data. Meanwhile, it maximizes the mutual information between $Z$ and the imputation

target $X$, to capture both fine-grained local details and global semantic features. This combination encourages the model to learn a well-aligned representation of the original data distribution, thereby addressing the aforementioned optimization dilemma and achieving accurate imputation.

## 3.2 Regularizing Partially Observed Input: $\min I_\theta(Z; X^o)$

Based on variational inference [50], we derive an upper bound for the regularization term in Eq. 3. The full derivation is shown in Appendix A.1.

$$
\begin{aligned}
I(Z; X^o) &= \int_{x^o} p(x^o) \cdot D_{\text{KL}}[p(z|x^o)||q(z)] \, dx^o - \int_{x^o} p(x^o|z) \cdot D_{\text{KL}}[p(z)||q(z)] \, dx^o, \\
&\leq \int_{x^o} p(x^o) \cdot D_{\text{KL}}[p(z|x^o)||q(z)] \, dx^o = \mathbb{E}_{p(x^o)} D_{\text{KL}}[p(z|x^o)||q(z)],
\end{aligned}
\tag{4}
$$

where the inequality follows from the non-negativity of KL divergence. Due to the difficulty in posterior calculation, we use our encoder $p_\theta(z \mid x^o)$ to approximate the true posterior distribution $p(z \mid x^o)$, so that the Regularization loss is defined as follows:

$$
I(Z; X^o) \leq \mathbb{E}_{p(x^o)} D_{\text{KL}}[p_\theta(z|x^o)||q(z)] \overset{\text{def}}{=} \mathcal{L}_{\text{Reg}}^\theta,
\tag{5}
$$

Meanwhile, we set an isotropic Gaussian as the prior distribution of the latent representations $Z$, i.e., $p(Z) = \mathcal{N}(0, I)$. Therefore, the encoder is defined to model partially observed time series data through a multivariate Gaussian distribution as shown below:

$$
p_\theta(Z|X^o) = \mathcal{N}(\mu_\theta(X^o), \text{diag}(\sigma_\theta(X^o))),
\tag{6}
$$

where $\mu_\theta(\cdot)$ and $\sigma_\theta(\cdot)$ are designed as neural networks with parameter $\theta$. During inference, we set the latent variable as $Z = \mu_\theta(X^o)$, and sample from the approximate posterior $Z \sim p_\theta(Z \mid X^o)$ using the reparameterization trick:

$$
Z = \mu_\theta(X^o) + \sigma_\theta(X^o) \odot \epsilon,
\tag{7}
$$

where $\epsilon \sim \mathcal{N}(0, I)$ and $\odot$ denote element-wise multiplication. Under this formulation, the Regularization loss in Eq. 5 can be computed and differentiated analytically as follows, without the need for stochastic estimation [23]:

$$
D_{\text{KL}} = \frac{1}{2} \sum_{j=1}^{d_{\text{model}}} \left( 1 + \log \left( \sigma_\theta^{(j)}(X^o) \right)^2 - \left( \mu_\theta^{(j)}(X^o) \right)^2 - \left( \sigma_\theta^{(j)}(X^o) \right)^2 \right).
\tag{8}
$$

Here, $d_{\text{model}}$ denotes the dimensionality of the latent representations, and $\mu_\theta^{(j)}(X^o)$ and $\sigma_\theta^{(j)}(X^o)$ represent the $j$-th elements of the mean and standard deviation vectors, respectively.

## 3.3 Maximizing Global and Local Inforamtion: $\max I_\phi(X; Z)$

**Local Mutual Information Maximization** Following the derivations introduced in previous IB-relevant literature [7, 1], we can obtain a lower bound for the informative term, which aims to maximize the mutual information between the latent representations $Z$ and the original data $X$ (full derivation is illustrated in Appendix A.2.1):

$$
\begin{aligned}
I(X; Z) &= \mathbb{E}_{p(x,z)} \left[ \log \frac{q_\phi(x|z)}{p(x)} \right] + \int_z p(z) \cdot D_{\text{KL}}[p(x|z)||q_\phi(x|z)] \, dz, \\
&\geq \mathbb{E}_{p(x,z)} \left[ \log q_\phi(x|z) \right] - \mathbb{E}_{p(x,z)} \left[ \log p(x) \right], \\
&\geq \mathbb{E}_{p(x,z)} \left[ \log q_\phi(x|z) \right] \overset{\text{def}}{=} -\mathcal{L}_{\text{Loc}}^\phi,
\end{aligned}
\tag{9}
$$

where the inequality holds due to the non-negativity of KL divergence and entropy. As we assume that time series data follow a Gaussian distribution with fixed variance [7, 23], i.e, $q_\phi(x|z) = \mathcal{N}(\hat{x}, \sigma^2 I)$, the derived Local loss can be further reduced to the form of a MSE loss as follows:

$$
\begin{aligned}
\mathcal{L}_{\text{Loc}}^\phi &= -\mathbb{E}_{p(x,z)} \left[ \log q_\phi(x|z) \right] = \mathbb{E}_{p(x,z)} \left[ \frac{1}{2\sigma^2} \|x - \hat{x}\|^2 + \frac{T}{2} \log(2\pi\sigma^2) \right], \\
&\propto \mathbb{E}_{p(x,z)} \left[ \|x - \hat{x}\|^2 \right],
\end{aligned}
\tag{10}
$$

where $\hat{x}$ denotes the imputation results generated by the model, and $T$ is the length of the time series. However, although this MSE-based Local loss provides a valid way to maximize $I(X; Z)$, it inherently emphasizes accurate reconstruction of local numerical values. These values often contain noise introduced by data collection errors and provide little guidance at the global level. Consequently, under high missing rates, the model tends to memorize these noisy details rather than learn the true data distribution, leading to poor generalization, degraded imputation quality, and severe distortion in the latent space. We identify this noise memorization as the key reason why both non-IB and IB-based TSI methods fail in such settings.

**Global Mutual Information Maximization** To overcome the limitations of point-wise reconstruction losses, we introduce a complementary formulation that explicitly targets the global (semantic-level) mutual information between the latent representations $Z$ and the original data $X$. Inspired by the InfoNCE objective from contrastive learning [38], we derive an alternative lower bound of $I(X; Z)$ (full derivation is illustrated in Appendix A.2.2):

$$
I(X; Z) = -\mathbb{E}_{p(x,z)}\left[\log\left(\frac{p(x)}{p(x|z)} \cdot N\right) - \log N\right] \approx -\mathbb{E}_{p(x,z)}\left[\log\left(\frac{p(x)}{p(x|z)} \cdot N\right)\right],
$$

$$
\geq \mathbb{E}_{p(x,z)}\left[\log\left(\frac{\frac{p(x|z)}{p(x)}}{\frac{p(x|z)}{p(x)} + \sum\limits_{x_j \in X^{\text{neg}}} \frac{p(x_j|z)}{p(x_j)}}\right)\right]. \tag{11}
$$

Instead of reconstructing $x$ directly with a decoder $q_\phi(x|z)$, we model a density ratio $f(x, z) = \exp(\text{proj}(z)^\top \cdot p_\theta(x))$ that preserves mutual information $I(X; Z)$, as it is proportional to $\frac{p(x|z)}{p(x)}$. And we denote $Z' = p_\theta(x)$. This yields the following Global Alignment loss:

$$
I(X; Z) \geq \mathbb{E}_{p(x,z)}\left[\log\left(\frac{f(x, z)}{f(x, z) + \sum\limits_{x_j \in X^{\text{neg}}} f(x_j, z)}\right)\right] \overset{\text{def}}{=} -\mathcal{L}_{\text{Glo\_1}}^\phi, \tag{12}
$$

where $\text{proj}(\cdot)$ and $p_\theta(\cdot)$ are a simple one-layer MLP and the model's encoder, respectively. Since our goal is to maximize global semantic-level mutual information, we treat the embedding of the original data at the same timestamp as the positive sample for the partially observed input. For negatives, we use embeddings from other timestamps of the original data. This setup pushes the model to align partially observed inputs with their original counterparts, encouraging it to capture semantic-level features such as temporal dynamics and global data distribution.

Moreover, considering the evolution of the training paradigm of the contrastive learning [15, 5], we can further simplify the Global Alignment loss $\mathcal{L}_{\text{Glo\_1}}^\phi$ to a simple alignment loss as follows:

$$
\mathcal{L}_{\text{Glo\_1}}^\phi \approx -\mathbb{E}_{p(x,z)}[f(x, z)] = \mathbb{E}_{p(x,z)}\left[\exp\left(\text{proj}(z)^\top \cdot \text{enc}(x)\right)\right] \overset{\text{def}}{=} \mathcal{L}_{\text{Glo\_2}}^\phi. \tag{13}
$$

### 3.4 Overall Training Objective

We now present the overall training objective of our proposed training paradigm **Glocal-IB**. This framework is simple to apply to any encoder-decoder architecture and requires only one additional MLP. By combining all components, including Regularization loss $\mathcal{L}_{\text{Reg}}^\theta$, Local loss $\mathcal{L}_{\text{Loc}}^\phi$, and Global Alignment loss $\mathcal{L}_{\text{Glo}}^\phi$, we optimize the time series imputation objective defined in Eq. 3:

$$
\min_{\theta, \phi}\left[\alpha \cdot \mathcal{L}_{\text{Reg}}^\theta + \beta_1 \cdot \mathcal{L}_{\text{Loc}}^\phi + \beta_2 \cdot \mathcal{L}_{\text{Glo}}^\phi\right], \tag{14}
$$

where $\alpha$, $\beta_1$, and $\beta_2$ are hyper-parameters that balance the mutual information. Global Alignment loss $\mathcal{L}_{\text{Glo}}^\phi$ can be implemented by $\mathcal{L}_{\text{Glo\_1}}^\phi$ or $\mathcal{L}_{\text{Glo\_2}}^\phi$.

## 4 Experiments

### 4.1 Experimental Settings

**Datasets**: Comprehensive experiments are conducted on nine public time-series datasets [59, 75, 26, 73], including ETTh1, ETTh2, ETTm1, ETTm2, Beijing Air, PEMS-Traffic, Electricity, Weather, and

Table 1: Imputation performance on 9 datasets (average MAE and MSE across 10% to 90% missing rates). Best is **bold** and second-best is underlined. We use OOM to denote out of memory.

| Models | IMP | Ours | | SAITS | | Transformer | | DLinear | | TimesNet | | FreTS | | PatchTST | | iTransformer | | GPVAE | | TimeMixer | |
|---|---|---|---|---|---|---|---|---|---|---|---|---|---|---|---|---|---|---|---|---|---|
| Metric | MSE | MAE | MSE | MAE | MSE | MAE | MSE | MAE | MSE | MAE | MSE | MAE | MSE | MAE | MSE | MAE | MSE | MAE | MSE | MAE | MSE |
| ETTh1 | 38% | **0.283** | **0.197** | 0.402 | 0.376 | 0.399 | 0.373 | 0.390 | 0.316 | 0.602 | 0.702 | 0.446 | 0.394 | 0.624 | 0.780 | 0.441 | 0.406 | 0.731 | 0.928 | 0.678 | 0.886 |
| ETTh2 | 40% | **0.249** | **0.132** | 0.340 | 0.256 | 0.307 | 0.218 | 0.352 | 0.243 | 0.800 | 1.140 | 0.434 | 0.370 | 0.525 | 0.575 | 0.413 | 0.321 | 0.686 | 0.769 | 0.529 | 0.535 |
| ETTm1 | 28% | **0.157** | **0.069** | 0.206 | 0.099 | 0.202 | 0.096 | 0.284 | 0.172 | 0.789 | 1.087 | 0.310 | 0.195 | 0.294 | 0.188 | 0.315 | 0.208 | 0.588 | 0.627 | 0.359 | 0.242 |
| ETTm2 | 25% | **0.157** | **0.069** | 0.206 | 0.099 | 0.202 | 0.096 | 0.284 | 0.172 | 0.789 | 1.087 | 0.310 | 0.195 | 0.294 | 0.188 | 0.315 | 0.208 | 0.588 | 0.627 | 0.359 | 0.242 |
| Beijing Air | 7% | **0.223** | **0.320** | 0.256 | 0.353 | 0.268 | 0.375 | 0.279 | 0.338 | 0.264 | 0.370 | 0.289 | 0.349 | 0.365 | 0.472 | 0.365 | 0.473 | 0.380 | 0.483 | 0.448 | 0.628 |
| PEMS-Traffic | 6% | **0.318** | **0.630** | 0.336 | 0.674 | 0.355 | 0.695 | 0.401 | 0.696 | 0.336 | 0.683 | 0.441 | 0.745 | 0.472 | 0.870 | OOM | OOM | 0.383 | 0.680 | 0.528 | 1.018 |
| Electricity | 8% | **0.372** | **0.296** | 0.397 | 0.343 | 0.410 | 0.358 | 0.483 | 0.433 | 0.390 | 0.322 | 0.544 | 0.534 | 0.676 | 0.783 | 0.440 | 0.363 | 0.443 | 0.394 | 0.648 | 0.724 |
| Weather | 34% | **0.096** | **0.056** | 0.136 | 0.093 | 0.139 | 0.091 | 0.161 | 0.089 | 0.262 | 0.211 | 0.167 | 0.085 | 0.188 | 0.111 | 0.182 | 0.102 | 0.278 | 0.195 | 0.239 | 0.180 |
| Metr-LA | 17% | **0.267** | **0.293** | 0.301 | 0.392 | 0.306 | 0.387 | 0.387 | 0.412 | 0.289 | 0.354 | 0.414 | 0.453 | 0.423 | 0.544 | 0.427 | 0.477 | 0.420 | 0.463 | 0.607 | 1.000 |

Metr-LA. During the experiments, we follow the point-wise missing patterns to randomly mask the time series [11]. We follow the standard train/validation/test splits provided by PyPOTS[1] [9]. More details are shown in the Appendix B.

**Baselines**: We select nine representative time series methods as our baselines, including: (1) Transformer-based methods: SAITS [10], Transformer [48], PatchTST [37], iTransformer [28]; (2) Linear-based methods: DLinear [71], FreTS [68], TimeMixer [55]; (3) Generative-based method: GPVAE [14]; and (4) CNN-based method: TimesNet [60].

**Evaluation Metrics**: Following previous studies [62, 63], we utilize MAE and MSE to evaluate the imputation performance by measuring feature-wise imputation quality. Lower values indicate better.

**Implementation Details**: To demonstrate the effectiveness of Glocal-IB, we apply it to a vanilla 2-layer Transformer. This simple backbone, equipped with our training strategy, serves as our demonstration model and is compared against all baselines. More information is in Appendix C.

## 4.2 Overall Comparison

We comprehensively compare the imputation performance of different methods over 9 datasets with various missing rates and visualize the latent representation distributions of SAITS, TimesNet, and our proposed method, which are the best three TSI methods. Due to space limits, we report the average imputation results over five missing rates (0.1, 0.3, 0.5, 0.7, and 0.9) in Table 1. Full results are provided in Appendix D.1. Based on the comparison results, we summarize our observations (**Obs.**):

**Obs. ❶: Glocal-IB demonstrates superior performance improvement in TSI tasks.** As shown in Table 1 and 2, Glocal-IB achieves the lowest MAE and MSE across all 9 datasets, with several cases showing a substantial margin. Notably, on ETTh1, ETTh2, ETTm1, and ETTm2, Glocal-IB shows substantial reductions in MSE (up to 40%) compared to all baselines. Even on more challenging real-world datasets like Beijing Air, PEMS-Traffic, Electricity, and Metr-LA, which contain complex temporal patterns and noise that the vanilla Transformer is not good at processing, Glocal-IB helps the Transformer to surpass SAITS and TimesNet by non-trivial margins in both MAE and MSE. Moreover, on the Weather dataset, Glocal-IB outperforms the second-best method by a large margin, reflecting strong robustness to seasonal patterns.

**Obs. ❷: The distortions of the latent representation distribution can be well solved by Glocal-IB.** As shown in the Fig. 8, existing representative TSI methods produce increasingly distorted latent distributions as the missing rate increases. These distortions suggest that the models fail to preserve the underlying temporal or structural properties of the original data under high missingness. In contrast, our proposed Glocal-IB maintains a stable and coherent latent structure from 10% to 70% missing rates. Even at 90% missingness, Glocal-IB still enables the model to capture the global

---
[1] https://github.com/WenjieDu/PyPOTS

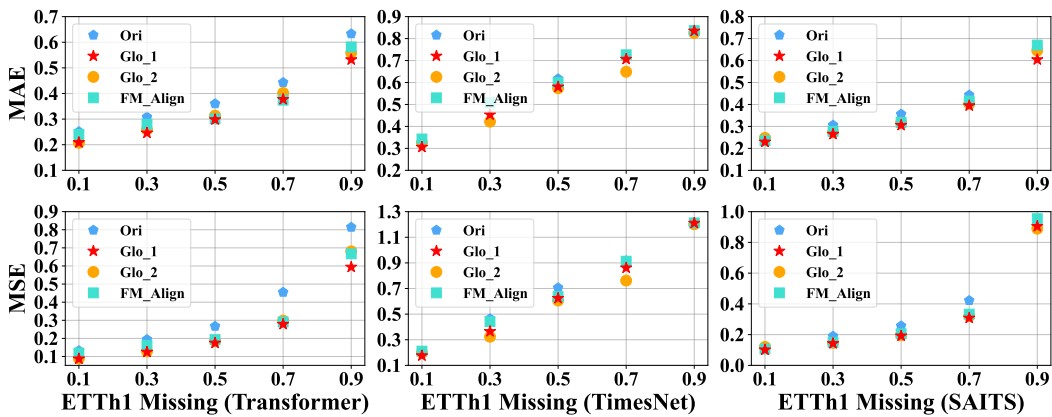

Figure 3: Imputation performance on the ETTh1 dataset of Transformer, TimesNet, and SAITS with four different training methods.

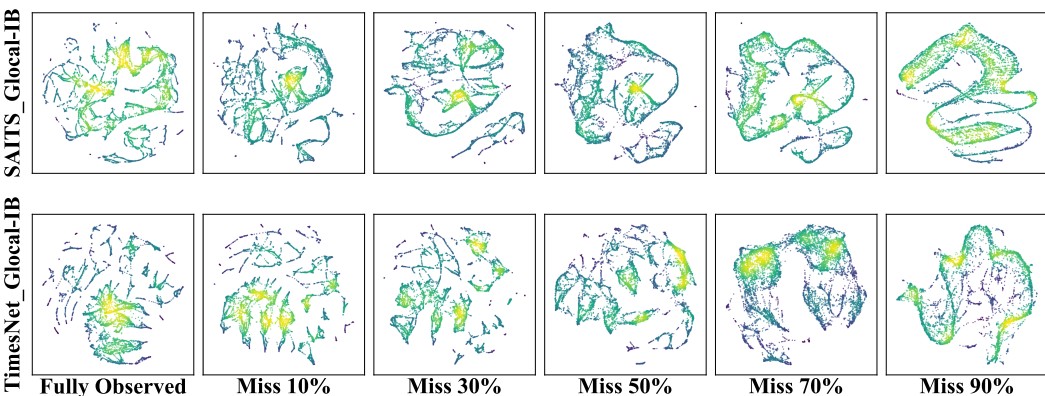

Figure 4: Latent space of SAITS and TimesNet with Glocal-IB on the ETTh1 dataset. Comparison with original models is in the Appendix D.1.

shape of the original distribution, while other methods show significant collapse or fragmentation in the latent space. This observation supports that Glocal-IB preserves informative global-local dependencies under extreme data degradation.

## 4.3 Generality Analysis

We conduct a series of studies to investigate how different training paradigms affect the performance of TSI models. Specifically, we select two of the most effective TSI methods—TimesNet and SAITS—and evaluate them under four training paradigms: (**1**) Ori: Standard reconstruction-based training without any external alignment. (**2**) FM_align: Representation alignment with a time series foundation model, specifically using the latest Time-MoE [45]. (**3**) Glo_1: Employ Eq. 12 as the usage of Global Alignment loss $\mathcal{L}_{\text{Glo}}^{\phi}$. And (**4**) Glo_2: Employ Eq. 13 as the usage of Global Alignment loss $\mathcal{L}_{\text{Glo}}^{\phi}$ to compare with Glo_1. From Fig. 3 and 4, we observe that:

**Obs. ❸: Glocal-IB improves the learning capability of existing imputation models.** From 10% to 70% missing, models trained with Glo_1 or Glo_2 consistently outperform the original version of baselines (Ori) and foundation model-aligned (FM_align) counterparts. Even under an extreme missing rate of 90%, where global information is little in only 10% observed data, Glocal-IB continues to enhance performance for both Transformer and SAITS, highlighting its effectiveness to boost TSI methods. Importantly, this improvement is achieved with minimal architectural modifications—only a lightweight MLP is introduced.

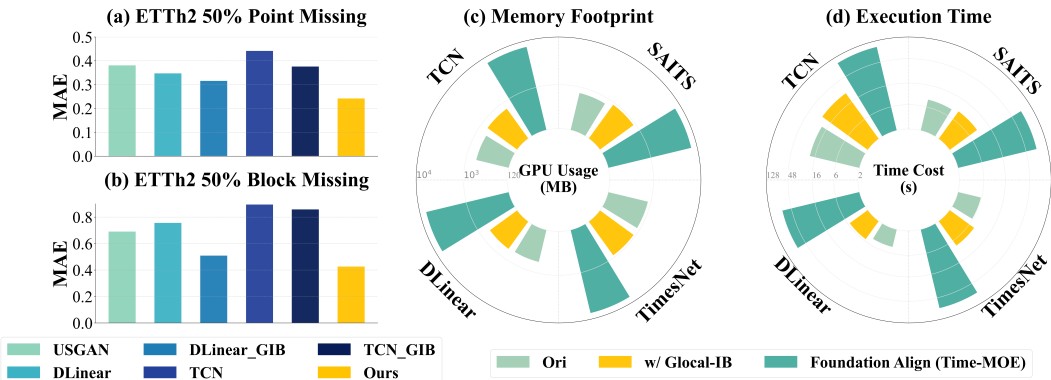

Figure 5: Different Missing Pattern Imputation and Efficiency results. **(a, b)**: Comparison of imputation performance on the ETTh2 dataset with 50% Point and Block missing rates. Additional results for various missing rates are presented in Appendix D.3. **(c, d)**: Efficiency comparison of four representative models on the ETTh1 dataset, evaluating the original models against their variants with Glocal-IB and foundation model alignment (Time-MoE). The radial axes are on a **logarithmic** scale.

**Obs. ❹: The Time series foundation model provides limited benefit.** We observe that Time-MoE-based alignment yields only marginal improvements for TSI tasks. This discrepancy is likely due to the nature of the pretraining objectives used in current time series foundation models [45, 29, 54], which are predominantly forecasting tasks. Such tasks may not impose sufficient semantic constraints on the learned representations, thereby limiting the benefit of alignment when transferred to imputation.

**Obs. ❺: Glocal-IB mitigates latent representation distortion.** Figure 1 and 4 illustrate the impact of Glocal-IB on latent representations. In the original SAITS and TimesNet model, the upper portion of the latent space becomes increasingly distorted as the missing rate grows from 10% to 30%. From 50% to 90% missing, the latent distribution collapses, indicating that the model fails to capture meaningful structure. In contrast, when they are trained with Glocal-IB, the latent distributions remain well-structured up to a 90% missing rate. This indicates that Glocal-IB introduces strong global regularization, enabling the model to preserve semantic coherence even under severe missingness.

## 4.4 Missing Pattern and Efficiency Analysis

We conduct experiments to analyze the effectiveness of Glocal-IB under various missing patterns and its efficiency. Our evaluation includes a suite of representative baselines: USGAN [33], DLinear [71], TCN [2], SAITS [10], and TimesNet [60]. Based on the results, we have the following observations:

**Obs. ❻: Our proposed method remains highly effective even for challenging block-wise missing patterns.** Figure 5 (b) shows that when contiguous blocks of data are missing—a scenario that disrupts local temporal dependencies—our method still achieves the lowest MAE by a significant margin. This demonstrates its robustness and superior capability in reconstructing structured data loss compared to other baselines.

**Obs. ❼: Glocal-IB enhances the capability of existing models across different missing patterns.** As shown in Figures 5 (a,b), applying Glocal-IB on current methods (e.g., DLinear_GIB and TCN_GIB) leads to improved imputation accuracy over the base models. This enhancement is particularly significant in the more challenging Block Missing scenario, where both DLinear_GIB and TCN_GIB achieve a lower MAE. This demonstrates Glocal-IB's broad utility in strengthening existing imputation methods.

**Obs. ❽: Glocal-IB is a computationally efficient module.** Figures 5 (c,d) reveal that augmenting existing models with our proposed Glocal-IB (w/ Glocal-IB) results in only a marginal increase in memory footprint and execution time compared to the original (Ori) versions. This efficiency stands in stark contrast to the Foundation Align based on the Time-MOE [45] method, which incurs substantial computational overhead. This highlights Glocal-IB as a practical, lightweight solution for enhancing model performance.

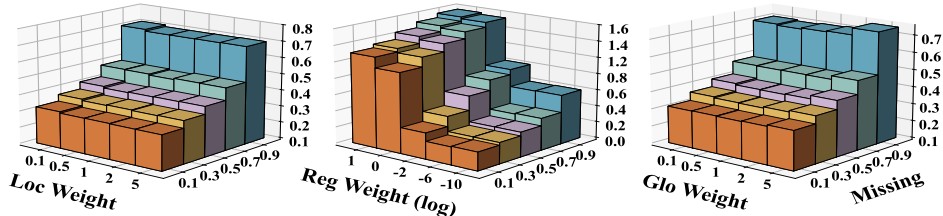

Figure 6: Hyperparameter sensitivity experiment results. More results are in Appendix D.4.

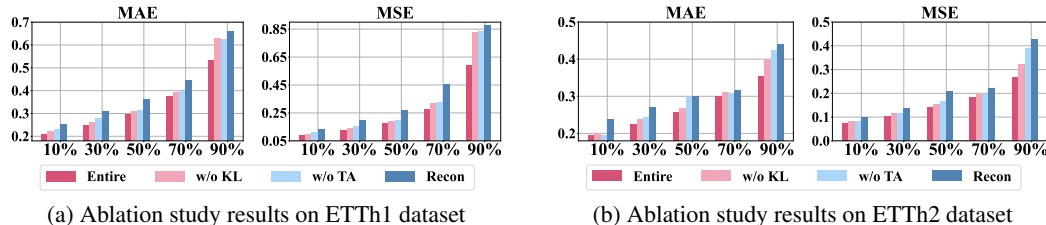

(a) Ablation study results on ETTh1 dataset      (b) Ablation study results on ETTh2 dataset

Figure 7: Visualization of ablation study results on ETTh1 dataset. More results are in Appendix D.4.

## 4.5 Ablation Study and Sensitivity Analysis

We conduct an ablation study and a parameter sensitivity analysis to examine the contribution and robustness of each component in Glocal-IB. The experiments are performed on four datasets: ETTh1, ETTh2, ETTm1, and ETTm2. In the **Ablation Study** (Fig. 7a and 7b), we compare the following configurations: (1) **Entire**: the full Glocal-IB method. (2) **w/o Reg**: Glocal-IB without the mutual information minimization term, Regularization loss. (3) **w/o Glo**: Glocal-IB without the global mutual information maximization, Global Alignment loss. (4) **only Loc**: only the reconstruction objective, i.e., Local loss, is used, corresponding to local mutual information maximization. In the **Sensitivity Analysis** (Fig. 6), we vary the weights assigned to the Local loss $\mathcal{L}_{\text{Glo}}^{\phi}$, Regularization loss $\mathcal{L}_{\text{Reg}}^{\theta}$, and Global Alignment loss $\mathcal{L}_{\text{Loc}}^{\phi}$ to study how each impacts model performance.

**Obs. ➒: Both the $\mathcal{L}_{\text{Reg}}^{\theta}$ and $\mathcal{L}_{\text{Glo}}^{\phi}$ are critical for improving imputation quality.** As demonstrated in Fig. 7, when either the Regularization loss $\mathcal{L}_{\text{Reg}}^{\theta}$ or the Global Alignment loss $\mathcal{L}_{\text{Glo}}^{\phi}$ is removed, the model performance deteriorates more significantly as the missing rate increases. This indicates that the $\mathcal{L}_{\text{Reg}}^{\theta}$ is effective at suppressing irrelevant variations in the latent space, while the $\mathcal{L}_{\text{Glo}}^{\phi}$ helps the model maintain global semantic information of the data.

**Obs. ➓: Imputation quality is sensitive to the weight of $\mathcal{L}_{\text{Reg}}^{\theta}$.** As shown in Fig. 6, increasing the weight of the Global Alignment loss $\mathcal{L}_{\text{Glo}}^{\phi}$ or Local loss $\mathcal{L}_{\text{Loc}}^{\phi}$ leads to stable performance trends. However, the imputation quality drops sharply when the weight of Regularization loss $\mathcal{L}_{\text{Reg}}^{\theta}$ exceeds 0.01. This suggests that a small amount of KL regularization is beneficial for filtering noise, but excessive regularization would suppress useful latent information excessively, resulting in significantly degraded imputation performance.

## 5 Conclusion

This paper studies the optimization dilemma in current TSI methods. To address this issue, we introduce a novel training paradigm, Glocal-IB. It extends standard IB-based objectives by adding a Global Alignment loss based on a tractable mutual information approximation. This loss encourages the latent representations of masked inputs to match those of their fully observed counterparts, helping the model retain global structure and local detail while reducing the impact of noise. Extensive experiments on nine datasets show that Glocal-IB consistently improves imputation accuracy and leads to more stable latent representation distributions under varying missing rates.

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

# A Theoretical Analysis

## A.1 Variational Approximation of Mutual Information Minimization for $\mathcal{L}_{\text{Reg}}^{\theta}$

Following prior work [1, 7], we approximate $I(Z; X^{\text{o}})$ using a variational upper bound. We begin by rewriting the mutual information definition into an equivalent KL form:

$$
\begin{aligned}
I(Z; X^{\text{o}}) &= \mathbb{E}_{p(z,x^{\text{o}})} \left[ \log \frac{p(z, x^{\text{o}})}{p(z) \cdot p(x^{\text{o}})} \right], \\
&= \mathbb{E}_{p(z,x^{\text{o}})} \left[ \log \frac{p(z|x^{\text{o}}) \cdot p(x^{\text{o}})}{p(z) \cdot p(x^{\text{o}})} \right], \\
&= \mathbb{E}_{p(z,x^{\text{o}})} \left[ \log \frac{p(z|x^{\text{o}})}{p(z)} \right].
\end{aligned}
\tag{15}
$$

To make the equation tractable, we follow the variational inference by introducing a variational marginal $q(z)$. We can convert $I(Z; X^{\text{o}})$ as follows:

$$
\begin{aligned}
I(Z; X^{\text{o}}) &= \mathbb{E}_{p(z,x^{\text{o}})} \left[ \log \frac{p(z|x^{\text{o}})}{p(z)} \right], \\
&= \mathbb{E}_{p(z,x^{\text{o}})} \left[ \log \left( \frac{p(z|x^{\text{o}})}{p(z)} \cdot \frac{q(z)}{q(z)} \right) \right], \\
&= \mathbb{E}_{p(z,x^{\text{o}})} \left[ \log \frac{p(z|x^{\text{o}})}{q(z)} - \log \frac{p(z)}{q(z)} \right], \\
&= \int_{x^{\text{o}}} \int_z \left[ p(z, x^{\text{o}}) \cdot \log \frac{p(z|x^{\text{o}})}{q(z)} - p(z, x^{\text{o}}) \cdot \log \frac{p(z)}{q(z)} \right] dz \, dx^{\text{o}}, \\
&= \int_{x^{\text{o}}} \int_z \left[ p(x^{\text{o}}) \cdot p(z|x^{\text{o}}) \cdot \log \frac{p(z|x^{\text{o}})}{q(z)} - p(x^{\text{o}}|z) \cdot p(z) \cdot \log \frac{p(z)}{q(z)} \right] dz \, dx^{\text{o}},
\end{aligned}
\tag{16}
$$

where the first and second terms are both calculations of KL divergence, so we can get as follows:

$$
\begin{aligned}
I(Z; X^{\text{o}}) &= \int_{x^{\text{o}}} \int_z \left[ p(x^{\text{o}}) \cdot p(z|x^{\text{o}}) \cdot \log \frac{p(z|x^{\text{o}})}{q(z)} - p(x^{\text{o}}|z) \cdot p(z) \cdot \log \frac{p(z)}{q(z)} \right] dz \, dx^{\text{o}}, \\
&= \int_{x^{\text{o}}} p(x^{\text{o}}) \cdot D_{\text{KL}}[p(z|x^{\text{o}})||q(z)] \cdot dx^{\text{o}} - \int_{x^{\text{o}}} p(x^{\text{o}}|z) \cdot D_{\text{KL}}[p(z)||q(z)] \, dx^{\text{o}}, \\
&\leq \int_{x^{\text{o}}} p(x^{\text{o}}) \cdot D_{\text{KL}}[p(z|x^{\text{o}})||q(z)] \, dx^{\text{o}}, \\
&= \mathbb{E}_{p(x^{\text{o}})} D_{\text{KL}}[p(z|x^{\text{o}})||q(z)],
\end{aligned}
\tag{17}
$$

where the last inequality follows from the non-negativity of KL divergence.

## A.2 Approximation of Mutual Information Maximization $\mathcal{L}_{\text{Loc}}^{\phi}$ and $\mathcal{L}_{\text{Glo}}^{\phi}$

### A.2.1 Derivation of $\mathcal{L}_{\text{Loc}}^{\phi}$

Here, we illustrate the entire derivation of the mutual information $I(X; Z)$ as in Eq. 9. Similar to the calculation in Eq. 15, by utilizing variational inference, we can get a lower bound of $I(X; Z)$:

$$
\begin{aligned}
I(X; Z) &= \mathbb{E}_{p(x,z)} \left[ \log \frac{p(x, z)}{p(x) \cdot p(z)} \right], \\
&= \mathbb{E}_{p(x,z)} \left[ \log \frac{p(x|z) \cdot p(z)}{p(x) \cdot p(z)} \right], \\
&= \mathbb{E}_{p(x,z)} \left[ \log \frac{p(x|z)}{p(x)} \right], \\
&= \mathbb{E}_{p(x,z)} \left[ \log \frac{p(x|z) \cdot q_{\phi}(x|z)}{p(x) \cdot q_{\phi}(x|z)} \right], \\
&= \mathbb{E}_{p(x,z)} \left[ \log \frac{q_{\phi}(x|z)}{p(x)} \right] + \mathbb{E}_{p(x,z)} \left[ \log \frac{p(x|z)}{q_{\phi}(x|z)} \right].
\end{aligned}
\tag{18}
$$

Note that the second term can be calculated as a KL divergence, so we calculate as follows:

$$
\begin{aligned}
I(X;Z) &= \mathbb{E}_{p(x,z)}\left[\log \frac{q_\phi(x|z)}{p(x)}\right] + \int_z \int_x p(x,z) \cdot \log \frac{p(x|z)}{q_\phi(x|z)}\, dx\, dz, \\
&= \mathbb{E}_{p(x,z)}\left[\log \frac{q_\phi(x|z)}{p(x)}\right] + \int_z \int_x p(x|z) \cdot p(z) \cdot \log \frac{p(x|z)}{q_\phi(x|z)}\, dx\, dz, \qquad (19)\\
&= \mathbb{E}_{p(x,z)}\left[\log \frac{q_\phi(x|z)}{p(x)}\right] + \int_z p(z) \cdot D_{\mathrm{KL}}[p(x|z)||q_\phi(x|z)]\, dz.
\end{aligned}
$$

Finally, because of the non-negativity of KL divergence:

$$
\begin{aligned}
I(X;Z) &= \mathbb{E}_{p(x,z)}\left[\log \frac{q_\phi(x|z)}{p(x)}\right] + \int_z p(z) \cdot D_{\mathrm{KL}}[p(x|z)||q_\phi(x|z)]\, dz. \\
&\geq \mathbb{E}_{p(x,z)}\left[\log \frac{q_\phi(x|z)}{p(x)}\right], \qquad (20)\\
&= \mathbb{E}_{p(x,z)}\left[\log q_\phi(x|z)\right] - \mathbb{E}_{p(x,z)}\left[\log p(x)\right], \\
&\geq \mathbb{E}_{p(x,z)}\left[\log q_\phi(x|z)\right].
\end{aligned}
$$

### A.2.2 Derivation of $\mathcal{L}_{\mathrm{Glo}}^{\phi}$

To provide the model with global-level guidance, we approximate the $I(X;Z)$ into a contrastive form, which is similar to the CPC [38].

$$
\begin{aligned}
I(X;Z) &= \mathbb{E}_{p(x,z)}\left[\log\left(\frac{p(x,z)}{p(x) \cdot p(z)}\right)\right], \\
&= \mathbb{E}_{p(x,z)}\left[\log\left(\frac{p(x|z) \cdot p(z)}{p(x) \cdot p(z)}\right)\right], \\
&= \mathbb{E}_{p(x,z)}\left[\log\left(\frac{p(x|z)}{p(x)}\right)\right], \\
&= -\mathbb{E}_{p(x,z)}\left[\log\left(\frac{p(x)}{p(x|z)}\right)\right], \\
&= -\mathbb{E}_{p(x,z)}\left[\log\left(\frac{p(x)}{p(x|z)} \cdot N\right) - \log N\right], \\
&\approx -\mathbb{E}_{p(x,z)}\left[\log\left(\frac{p(x)}{p(x|z)} \cdot N\right)\right], \qquad (21)\\
&\geq -\mathbb{E}_{p(x,z)}\left[\log\left(1 + \frac{p(x)}{p(x|z)} \cdot (N-1) \cdot 1\right)\right], \\
&= -\mathbb{E}_{p(x,z)}\left[\log\left(1 + \frac{p(x)}{p(x|z)} \cdot (N-1) \cdot \mathbb{E}_{p(x_j)}\left(\frac{p(x_j|z)}{p(x_j)}\right)\right)\right], \\
&= \mathbb{E}_{p(x,z)}\left[\log\left(\frac{\frac{p(x|z)}{p(x)}}{\frac{p(x|z)}{p(x)} + \sum_{x_j \in X^{\mathrm{neg}}} \frac{p(x_j|z)}{p(x_j)}}\right)\right], \\
&= \mathbb{E}_{p(x,z)}\left[\log\left(\frac{f(x,z)}{f(x,z) + \sum_{x_j \in X^{\mathrm{neg}}} f(x_j,z)}\right)\right].
\end{aligned}
$$

Here, we use a mini-batch approach [7] that $X^{\mathrm{neg}}$ is chosen from other timestamps' data in the same mini-batch. And $f(x,z)$ is a density ratio that is proportional to $\frac{p(x|z)}{p(x)}$.

Moreover, inspired by the evolution of the contrastive learning [15, 5, 77], we further simplify the Eq. 21 as shown below:

$$
I(X;Z) \approx -\mathbb{E}_{p(x,z)}\left[f(x,z)\right]. \qquad (22)
$$

Therefore, we get a simpler alignment loss in Eq. 13.

# B  Datasets

We conduct experiments on 9 real-world datasets to evaluate the imputation performance. Now we describe the detailed information of these 9 datasets as follows:

- **ETT** [75] records 7 power-related factors from electricity transformers between 2016/07 and 2018/07. It includes four subsets: **ETTh1** and **ETTh2** are sampled hourly, while **ETTm1** and **ETTm2** are sampled every 15 minutes.
- **Beijing Air** [73] provides hourly air quality data from 12 monitoring stations in Beijing, collected from 2013/03/01 to 2017/02/08. Each station measures 11 variables, resulting in 132 combined features.
- **PEMS-Traffic** [59] contains hourly road occupancy rates from 862 sensors on San Francisco Bay area highways, spanning 2015/01 to 2016/02.
- **Electricity** [59] records hourly electricity usage of 321 clients from 2012 to 2014.
- **Weather** [59] includes 21 meteorological variables collected every 10 minutes at the Max Planck Biogeochemistry Institute throughout 2021.
- **Metr-LA** [26] captures traffic speeds every 5 minutes from 207 road sensors across Los Angeles County, covering the period from 2012/03 to 2012/06.

# C  Implementation Details

We follow the data processing and split protocol from PyPOTS [9]. The training, validation, and test sets are divided (60%, 20%, and 20%) in chronological order to avoid data leakage. For all datasets, the input sequence length is set to 96.

All experiments are implemented in PyTorch [40] 2.6.0 and run on a single NVIDIA 4090 GPU with 24GB memory. We use the Adam optimizer [22] with a learning rate of 0.001. The batch size is 64, and the number of training epochs is fixed to 30. The hidden dimension is set to 256.

All baseline models are built upon the PyPOTS [9] benchmark, where each model follows the settings from its original paper and official implementation. We report the average results over 5 different random seeds in this paper.

# D  Full Experiments

## D.1  Full Comparison Results

Table 2 provides a comprehensive comparison of Globcal-IB with a vanilla Transformer against baseline methods, with results of missing rate of 10%, 30%, 50%, 70%, and 90% listed separately. Additionally, we visualize the latent space of ours and three representative TSI methods in Fig. 8.

## D.2  Generality Analysis

Fig. 9 provides the latent space of TimesNet, SAITS, and their Glocal-IB counterparts, across missing rate 10%, 30%, 50%, 70%, and 90%. Glocal-IB remarkably improves the alignment of the latent space while the missing rate increases.

## D.3  Missing Pattern Analysis

Fig. 10, 11, 12, and 13 provide the entire performance comparison results on ETTh1, ETTh2, ETTm1, and ETTm2, across missing rate 10%, 30%, 50%, 70%, and 90%. These indicate that our proposed method achieves the best imputation performance while remarkably improving the imputation performance of current methods.

## D.4  Ablation Study and Sensitivity Analysis

Fig. 15 illustrates all the ablation studies on 4 datasets, including ETTh1, ETTh2, ETTm1, and ETTm2.

Fig. 14 demonstrates all the parameter sensitivity analyses on 4 datasets, including ETTh1, ETTh2, ETTm1, and ETTm2.

## E   Societal Impact Statement

Similar to previous TSI works [66, 3], the development of Glocal-IB has the potential to benefit a wide range of real-world applications. In healthcare, for example, improved imputation models can enhance the reliability of patient monitoring systems by recovering missing clinical measurements. This may support earlier diagnosis, enable timely interventions, and help reduce overall medical costs by facilitating more informed decision-making.

However, the deployment of advanced imputation techniques also introduces several risks. In sensitive domains such as surveillance, these models may reconstruct incomplete data in ways that raise privacy concerns, particularly if used to infer personal information without consent. Moreover, over-reliance on automated imputation may lead to overlooked errors, potentially resulting in biased or unreliable decisions in downstream tasks.

To mitigate these risks, it is important to establish clear guidelines for the ethical use of imputation models. This includes enforcing data protection regulations, ensuring transparency in model behavior, and incorporating fairness-aware validation protocols. Broadening access to such technologies and conducting regular audits can further promote responsible deployment and prevent unintended harm.

## F   Limitation and Discussion

Due to computational constraints, we only apply Glocal-IB to three representative backbones—TimesNet, SAITS, and Transformer—on four datasets from the ETT benchmark: ETTh1, ETTh2, ETTm1, and ETTm2. While these results are sufficient to demonstrate the effectiveness of our approach, future work can explore broader model families and larger-scale datasets to further validate generalization.

We also observe that the performance gain under extreme missingness (e.g., 90%) is less pronounced than at moderate levels (10%–70%). A possible reason is that in the inference phase when only a small portion of the input is available (e.g., 10%), the preserved global structure is too weak to offer meaningful alignment signals. In such cases, the model has limited capacity to distinguish signal from noise, resulting in less reliable imputation.

This limitation highlights an important future direction: how to enhance global guidance under limited observations. One possible solution is to incorporate stronger structural priors or pretrained knowledge to better inform the latent space.

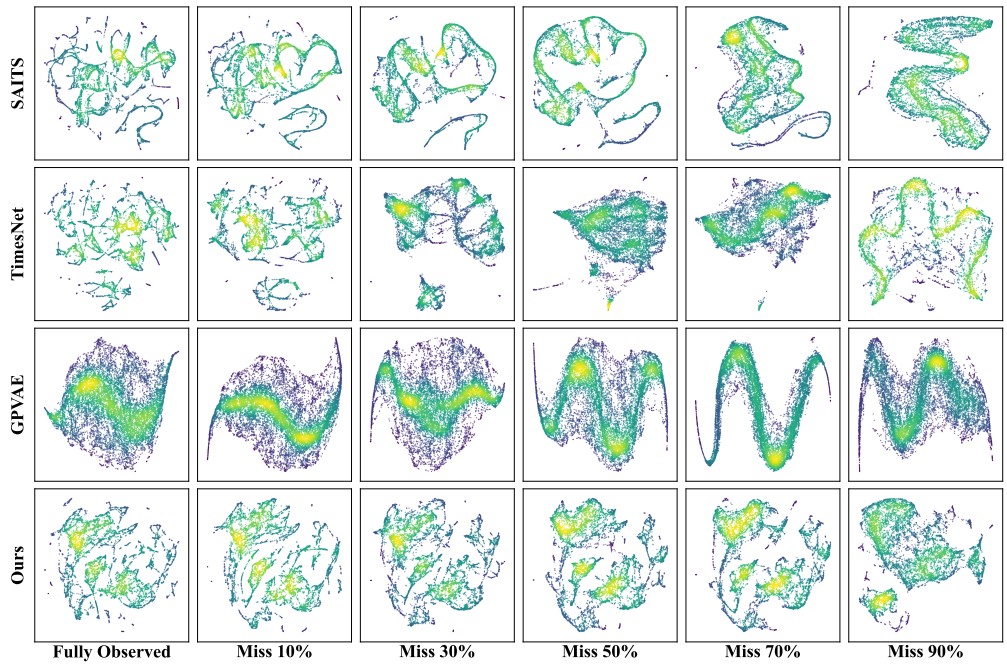

Figure 8: Latent space comparison of Glocal-IB on Transformer and three representative TSI methods, including SAITS, TimesNet, and GPVAE.

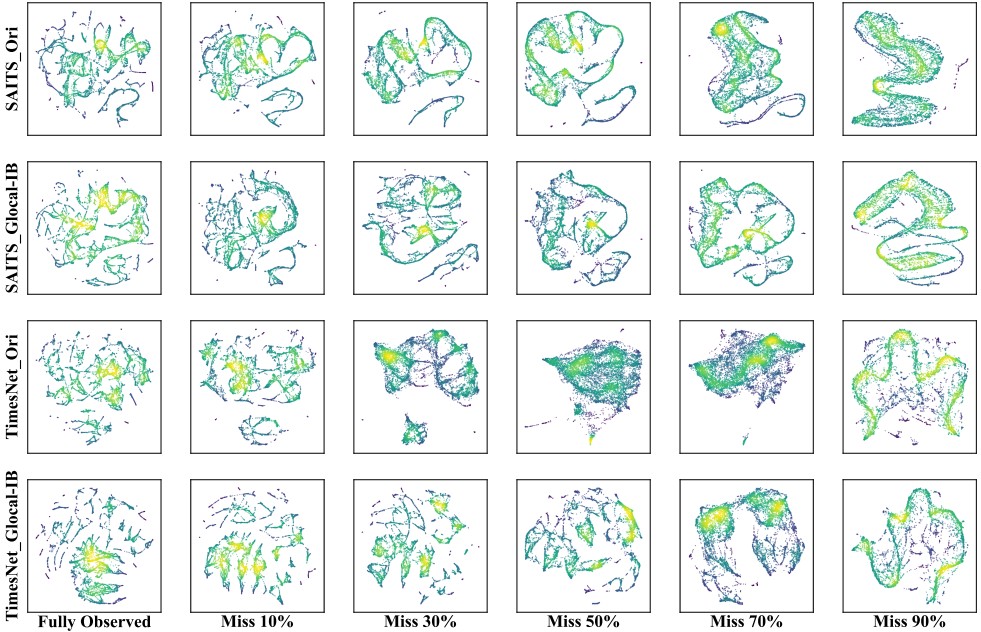

Figure 9: Latent space comparison of TimesNet and SAITS with Glocal-IB against their original implementations.

Table 2: Imputation performance on 9 datasets (average MAE and MSE across 10% to 90% missing rates). Best is **bold** and second-best is underlined. We use OOM to denote out of memory.

| Models | | IMP | Ours | | SAITS | | Transformer | | DLinear | | TimesNet | | FreTS | | PatchTST | | iTransformer | | GPVAE | | TimeMixer | |
|---|---|---|---|---|---|---|---|---|---|---|---|---|---|---|---|---|---|---|---|---|---|---|
| Metric | | MSE | MAE | MSE | MAE | MSE | MAE | MSE | MAE | MSE | MAE | MSE | MAE | MSE | MAE | MSE | MAE | MSE | MAE | MSE | MAE | MSE |
| ETTh1 | 0.1 | 50% | **0.165** | **0.060** | 0.245 | 0.121 | 0.251 | 0.132 | 0.281 | 0.158 | 0.329 | 0.209 | 0.324 | 0.191 | 0.494 | 0.480 | 0.350 | 0.246 | 0.664 | 0.736 | 0.626 | 0.730 |
| | 0.3 | 49% | **0.204** | **0.096** | 0.306 | 0.190 | 0.308 | 0.194 | 0.316 | 0.193 | 0.506 | 0.464 | 0.348 | 0.221 | 0.583 | 0.743 | 0.368 | 0.267 | 0.690 | 0.827 | 0.655 | 0.824 |
| | 0.5 | 47% | **0.239** | **0.128** | 0.356 | 0.257 | 0.360 | 0.267 | 0.356 | 0.244 | 0.617 | 0.706 | 0.384 | 0.267 | 0.546 | 0.614 | 0.401 | 0.314 | 0.735 | 0.901 | 0.683 | 0.925 |
| | 0.7 | 36% | **0.303** | **0.202** | 0.443 | 0.422 | 0.443 | 0.455 | 0.410 | 0.315 | 0.718 | 0.917 | 0.463 | 0.382 | 0.700 | 0.927 | 0.453 | 0.401 | 0.760 | 1.009 | 0.703 | 0.968 |
| | 0.9 | 26% | **0.506** | **0.497** | 0.658 | 0.890 | 0.633 | 0.814 | 0.585 | 0.670 | 0.839 | 1.216 | 0.712 | 0.909 | 0.796 | 1.134 | 0.633 | 0.799 | 0.808 | 1.166 | 0.721 | 0.983 |
| ETTh2 | 0.1 | 50% | **0.180** | **0.063** | 0.265 | 0.142 | 0.235 | 0.126 | 0.289 | 0.157 | 0.423 | 0.342 | 0.315 | 0.189 | 0.396 | 0.312 | 0.419 | 0.318 | 0.522 | 0.458 | 0.463 | 0.390 |
| | 0.3 | 48% | **0.205** | **0.080** | 0.276 | 0.167 | 0.260 | 0.155 | 0.317 | 0.192 | 0.756 | 0.955 | 0.383 | 0.271 | 0.423 | 0.347 | 0.387 | 0.282 | 0.545 | 0.496 | 0.465 | 0.383 |
| | 0.5 | 37% | **0.243** | **0.115** | 0.309 | 0.209 | 0.286 | 0.183 | 0.340 | 0.223 | 0.862 | 1.222 | 0.401 | 0.306 | 0.419 | 0.327 | 0.416 | 0.321 | 0.640 | 0.649 | 0.495 | 0.449 |
| | 0.7 | 44% | **0.254** | **0.132** | 0.350 | 0.256 | 0.324 | 0.235 | 0.363 | 0.262 | 0.957 | 1.511 | 0.456 | 0.381 | 0.525 | 0.520 | 0.381 | 0.267 | 0.764 | 0.889 | 0.531 | 0.549 |
| | 0.9 | 30% | **0.361** | **0.270** | 0.498 | 0.508 | 0.432 | 0.390 | 0.451 | 0.384 | 1.002 | 1.672 | 0.613 | 0.702 | 0.864 | 1.368 | 0.460 | 0.415 | 0.958 | 1.351 | 0.692 | 0.903 |
| ETTm1 | 0.1 | 45% | **0.097** | **0.024** | 0.138 | 0.042 | 0.145 | 0.047 | 0.215 | 0.095 | 0.707 | 0.876 | 0.245 | 0.119 | 0.218 | 0.094 | 0.243 | 0.123 | 0.469 | 0.377 | 0.318 | 0.187 |
| | 0.3 | 35% | **0.117** | **0.036** | 0.158 | 0.055 | 0.162 | 0.059 | 0.230 | 0.110 | 0.762 | 1.012 | 0.258 | 0.131 | 0.237 | 0.112 | 0.262 | 0.137 | 0.520 | 0.460 | 0.321 | 0.195 |
| | 0.5 | 35% | **0.135** | **0.049** | 0.186 | 0.075 | 0.185 | 0.075 | 0.251 | 0.129 | 0.799 | 1.112 | 0.279 | 0.151 | 0.248 | 0.122 | 0.294 | 0.168 | 0.546 | 0.514 | 0.335 | 0.208 |
| | 0.7 | 30% | **0.165** | **0.070** | 0.226 | 0.107 | 0.215 | 0.101 | 0.292 | 0.169 | 0.832 | 1.206 | 0.318 | 0.193 | 0.294 | 0.165 | 0.324 | 0.201 | 0.627 | 0.680 | 0.360 | 0.236 |
| | 0.9 | 15% | **0.271** | **0.167** | 0.324 | 0.217 | 0.304 | 0.197 | 0.430 | 0.358 | 0.844 | 1.231 | 0.452 | 0.381 | 0.473 | 0.445 | 0.450 | 0.411 | 0.777 | 1.102 | 0.463 | 0.381 |
| ETTm2 | 0.1 | 18% | **0.113** | **0.032** | 0.151 | 0.043 | 0.139 | 0.039 | 0.229 | 0.101 | 0.751 | 0.974 | 0.267 | 0.141 | 0.241 | 0.109 | 0.316 | 0.206 | 0.466 | 0.381 | 0.337 | 0.209 |
| | 0.3 | 25% | **0.126** | **0.039** | 0.160 | 0.052 | 0.169 | 0.056 | 0.253 | 0.129 | 0.866 | 1.263 | 0.287 | 0.155 | 0.305 | 0.177 | 0.317 | 0.207 | 0.479 | 0.407 | 0.319 | 0.193 |
| | 0.5 | 37% | **0.138** | **0.042** | 0.183 | 0.065 | 0.183 | 0.065 | 0.290 | 0.169 | 0.933 | 1.450 | 0.291 | 0.165 | 0.287 | 0.162 | 0.334 | 0.230 | 0.482 | 0.404 | 0.320 | 0.199 |
| | 0.7 | 18% | **0.169** | **0.063** | 0.231 | 0.102 | 0.199 | 0.078 | 0.305 | 0.188 | 0.978 | 1.588 | 0.337 | 0.222 | 0.288 | 0.162 | 0.330 | 0.220 | 0.498 | 0.439 | 0.363 | 0.265 |
| | 0.9 | 23% | **0.227** | **0.109** | 0.290 | 0.169 | 0.259 | 0.142 | 0.392 | 0.313 | 1.011 | 1.694 | 0.412 | 0.326 | 0.385 | 0.289 | 0.375 | 0.273 | 0.746 | 0.872 | 0.452 | 0.405 |
| Beijing Air | 0.1 | 9% | **0.192** | **0.289** | 0.216 | 0.291 | 0.231 | 0.344 | 0.253 | 0.294 | 0.231 | 0.359 | 0.272 | 0.316 | 0.320 | 0.382 | 0.298 | 0.362 | 0.339 | 0.422 | 0.440 | 0.606 |
| | 0.3 | 7% | **0.202** | **0.299** | 0.231 | 0.325 | 0.247 | 0.352 | 0.263 | 0.312 | 0.242 | 0.342 | 0.271 | 0.318 | 0.309 | 0.389 | 0.327 | 0.418 | 0.350 | 0.436 | 0.451 | 0.627 |
| | 0.5 | 6% | **0.215** | **0.312** | 0.250 | 0.343 | 0.260 | 0.363 | 0.269 | 0.336 | 0.244 | 0.337 | 0.276 | 0.336 | 0.324 | 0.407 | 0.347 | 0.445 | 0.371 | 0.471 | 0.435 | 0.602 |
| | 0.7 | 5% | **0.234** | **0.327** | 0.267 | 0.376 | 0.277 | 0.380 | 0.287 | 0.341 | 0.255 | 0.353 | 0.296 | 0.359 | 0.392 | 0.497 | 0.373 | 0.474 | 0.388 | 0.490 | 0.448 | 0.628 |
| | 0.9 | 5% | **0.272** | **0.374** | 0.317 | 0.431 | 0.324 | 0.436 | 0.322 | 0.406 | 0.348 | 0.462 | 0.333 | 0.417 | 0.478 | 0.686 | 0.480 | 0.666 | 0.454 | 0.597 | 0.466 | 0.674 |
| PEMS-Traffic | 0.1 | 9% | **0.303** | **0.602** | 0.330 | 0.664 | 0.347 | 0.685 | 0.394 | 0.668 | 0.333 | 0.681 | 0.455 | 0.770 | 0.440 | 0.783 | OOM | OOM | 0.388 | 0.680 | 0.522 | 0.984 |
| | 0.3 | 7% | **0.308** | **0.624** | 0.331 | 0.670 | 0.346 | 0.691 | 0.396 | 0.680 | 0.333 | 0.677 | 0.439 | 0.742 | 0.458 | 0.844 | OOM | OOM | 0.383 | 0.680 | 0.525 | 1.014 |
| | 0.5 | 6% | **0.318** | **0.630** | 0.334 | 0.674 | 0.349 | 0.691 | 0.393 | 0.678 | 0.333 | 0.688 | 0.439 | 0.740 | 0.462 | 0.855 | OOM | OOM | 0.374 | 0.671 | 0.526 | 0.998 |
| | 0.7 | 5% | **0.324** | **0.638** | 0.332 | 0.673 | 0.357 | 0.699 | 0.403 | 0.697 | 0.336 | 0.679 | 0.448 | 0.744 | 0.498 | 0.905 | OOM | OOM | 0.376 | 0.674 | 0.528 | 1.028 |
| | 0.9 | 5% | **0.339** | **0.657** | 0.354 | 0.691 | 0.375 | 0.710 | 0.417 | 0.757 | 0.345 | 0.693 | 0.425 | 0.728 | 0.500 | 0.962 | OOM | OOM | 0.394 | 0.693 | 0.536 | 1.067 |
| Electricity | 0.1 | 5% | **0.340** | **0.252** | 0.351 | 0.278 | 0.360 | 0.286 | 0.459 | 0.390 | 0.370 | 0.297 | 0.498 | 0.456 | 0.666 | 0.770 | 0.375 | 0.266 | 0.430 | 0.373 | 0.651 | 0.727 |
| | 0.3 | 7% | **0.349** | **0.261** | 0.355 | 0.282 | 0.366 | 0.294 | 0.465 | 0.399 | 0.373 | 0.300 | 0.571 | 0.582 | 0.696 | 0.811 | 0.393 | 0.289 | 0.430 | 0.373 | 0.655 | 0.739 |
| | 0.5 | 5% | **0.354** | **0.275** | 0.360 | 0.290 | 0.374 | 0.303 | 0.471 | 0.415 | 0.376 | 0.304 | 0.524 | 0.496 | 0.642 | 0.718 | 0.419 | 0.324 | 0.436 | 0.384 | 0.652 | 0.731 |
| | 0.7 | 4% | **0.379** | **0.310** | 0.435 | 0.393 | 0.455 | 0.422 | 0.483 | 0.434 | 0.392 | 0.324 | 0.579 | 0.597 | 0.694 | 0.812 | 0.456 | 0.378 | 0.447 | 0.401 | 0.643 | 0.712 |
| | 0.9 | 1% | **0.437** | **0.380** | 0.486 | 0.473 | 0.495 | 0.485 | 0.538 | 0.526 | 0.440 | 0.384 | 0.548 | 0.539 | 0.683 | 0.805 | 0.560 | 0.559 | 0.474 | 0.441 | 0.639 | 0.710 |
| Weather | 0.1 | 35% | **0.070** | **0.034** | 0.103 | 0.072 | 0.111 | 0.069 | 0.150 | 0.068 | 0.115 | 0.053 | 0.156 | 0.065 | 0.164 | 0.084 | 0.144 | 0.080 | 0.239 | 0.151 | 0.224 | 0.167 |
| | 0.3 | 41% | **0.073** | **0.039** | 0.110 | 0.075 | 0.118 | 0.076 | 0.150 | 0.073 | 0.135 | 0.075 | 0.149 | 0.066 | 0.157 | 0.087 | 0.158 | 0.083 | 0.252 | 0.161 | 0.235 | 0.170 |
| | 0.5 | 42% | **0.084** | **0.047** | 0.125 | 0.080 | 0.136 | 0.085 | 0.154 | 0.080 | 0.175 | 0.120 | 0.165 | 0.081 | 0.178 | 0.100 | 0.172 | 0.089 | 0.273 | 0.179 | 0.229 | 0.174 |
| | 0.7 | 32% | **0.103** | **0.059** | 0.143 | 0.091 | 0.142 | 0.093 | 0.159 | 0.093 | 0.336 | 0.269 | 0.165 | 0.087 | 0.204 | 0.118 | 0.199 | 0.110 | 0.265 | 0.180 | 0.235 | 0.175 |
| | 0.9 | 19% | **0.150** | **0.101** | 0.202 | 0.146 | 0.188 | 0.134 | 0.193 | 0.129 | 0.547 | 0.537 | 0.203 | 0.125 | 0.238 | 0.164 | 0.238 | 0.149 | 0.360 | 0.302 | 0.270 | 0.211 |
| Metr-LA | 0.1 | 23% | **0.247** | **0.250** | 0.281 | 0.350 | 0.290 | 0.353 | 0.381 | 0.383 | 0.267 | 0.323 | 0.409 | 0.423 | 0.388 | 0.459 | 0.383 | 0.381 | 0.409 | 0.436 | 0.646 | 1.045 |
| | 0.3 | 21% | **0.251** | **0.262** | 0.283 | 0.362 | 0.292 | 0.363 | 0.391 | 0.403 | 0.271 | 0.331 | 0.411 | 0.432 | 0.392 | 0.468 | 0.397 | 0.410 | 0.383 | 0.414 | 0.608 | 0.966 |
| | 0.5 | 20% | **0.259** | **0.277** | 0.292 | 0.375 | 0.298 | 0.372 | 0.380 | 0.397 | 0.279 | 0.346 | 0.439 | 0.509 | 0.420 | 0.509 | 0.420 | 0.464 | 0.380 | 0.411 | 0.605 | 1.003 |
| | 0.7 | 20% | **0.267** | **0.294** | 0.305 | 0.391 | 0.311 | 0.393 | 0.379 | 0.411 | 0.290 | 0.366 | 0.401 | 0.432 | 0.406 | 0.534 | 0.439 | 0.508 | 0.436 | 0.487 | 0.580 | 0.963 |
| | 0.9 | 5% | **0.308** | **0.383** | 0.345 | 0.482 | 0.342 | 0.456 | 0.406 | 0.468 | 0.336 | 0.404 | 0.412 | 0.472 | 0.509 | 0.750 | 0.498 | 0.624 | 0.494 | 0.567 | 0.597 | 1.025 |

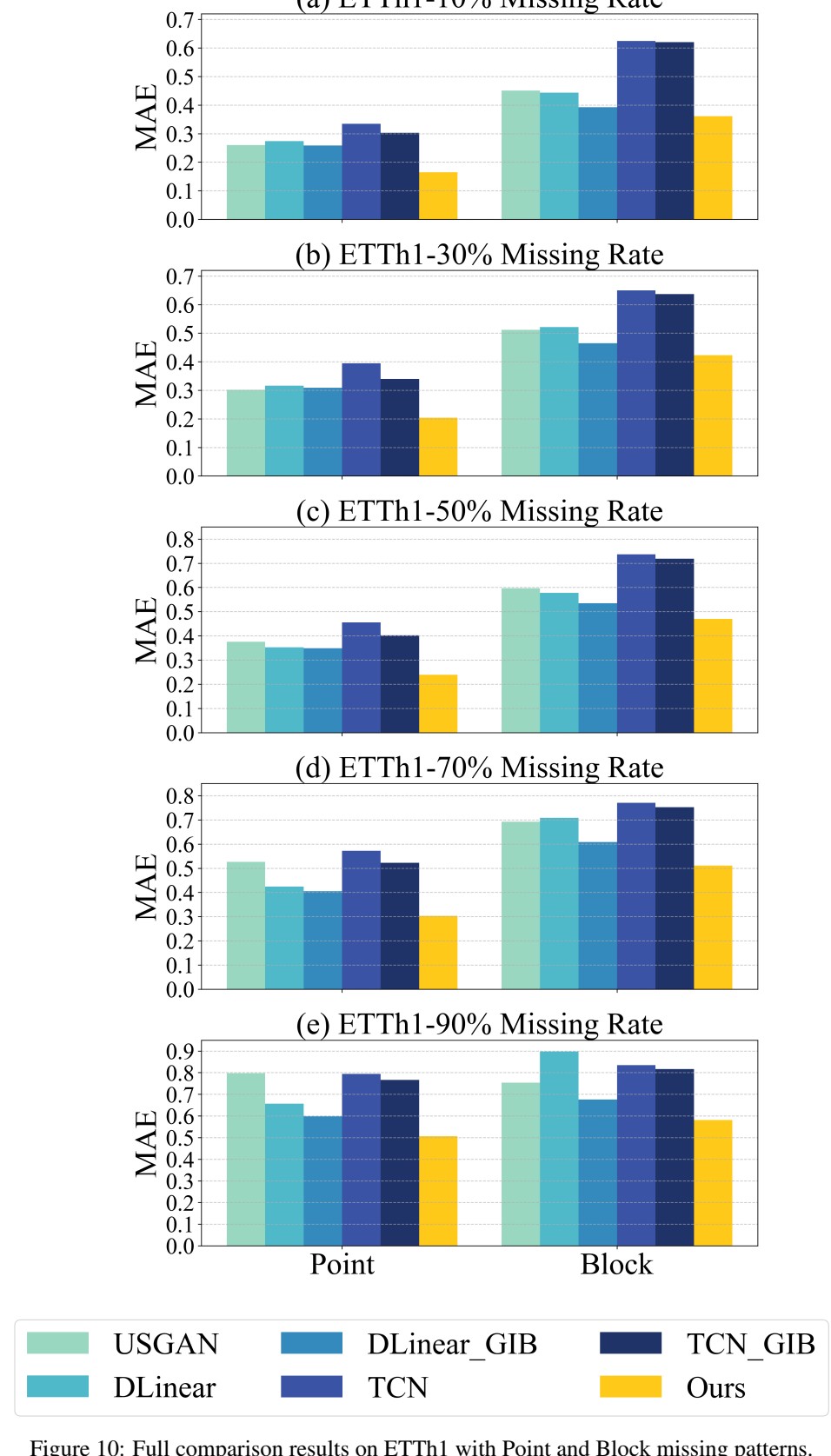

Figure 10: Full comparison results on ETTh1 with Point and Block missing patterns.

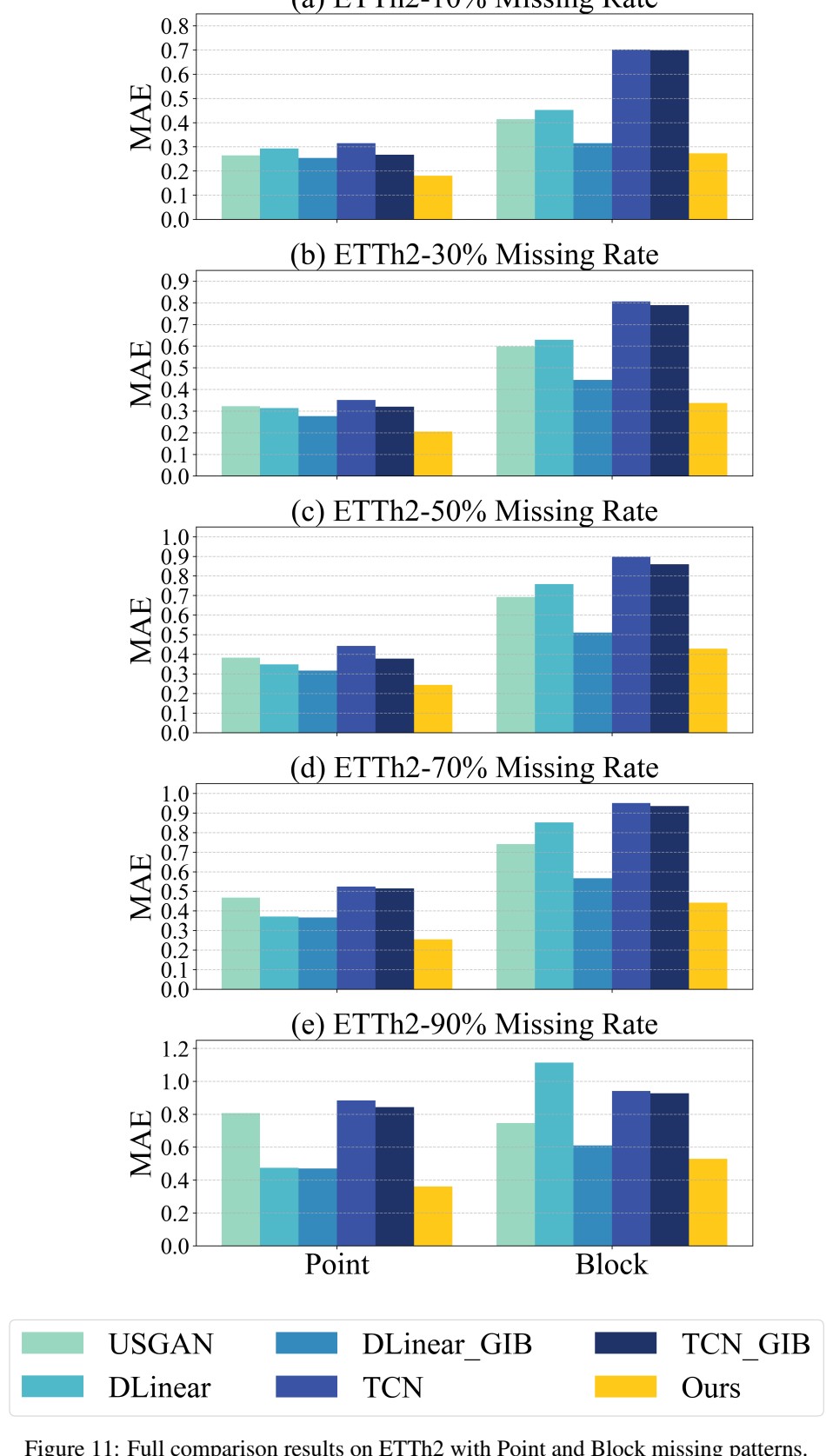

Figure 11: Full comparison results on ETTh2 with Point and Block missing patterns.

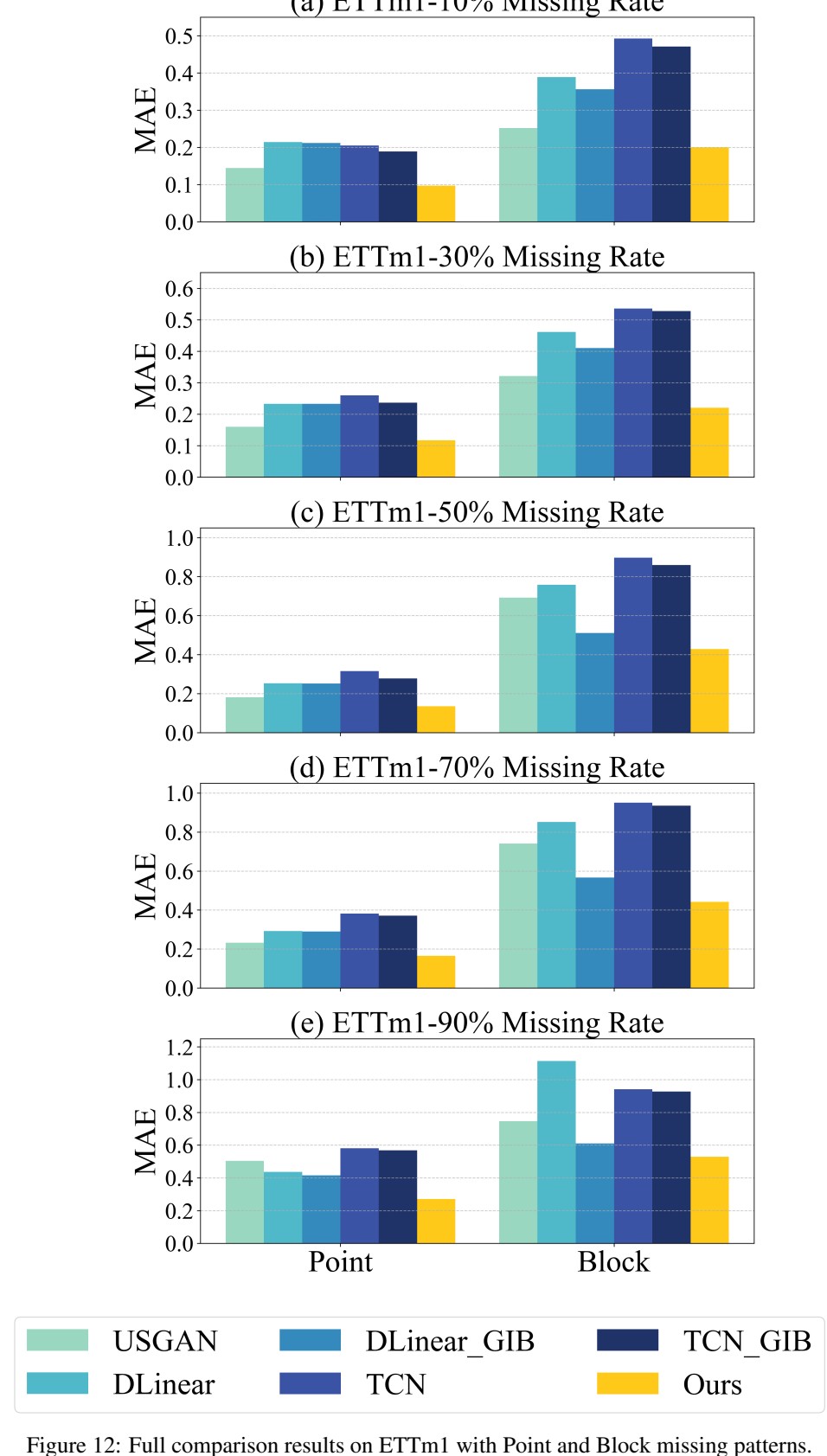

Figure 12: Full comparison results on ETTm1 with Point and Block missing patterns.

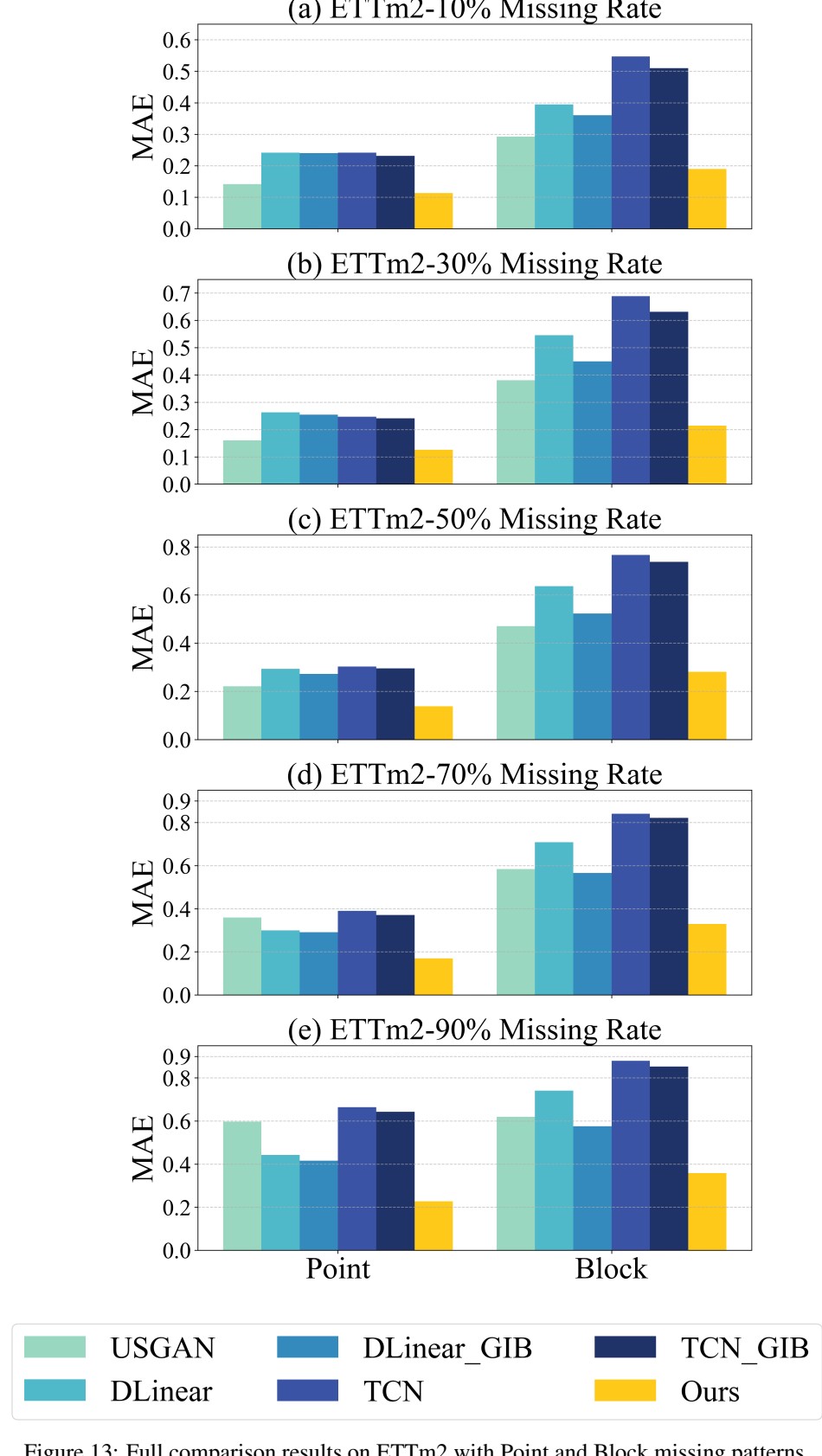

Figure 13: Full comparison results on ETTm2 with Point and Block missing patterns.

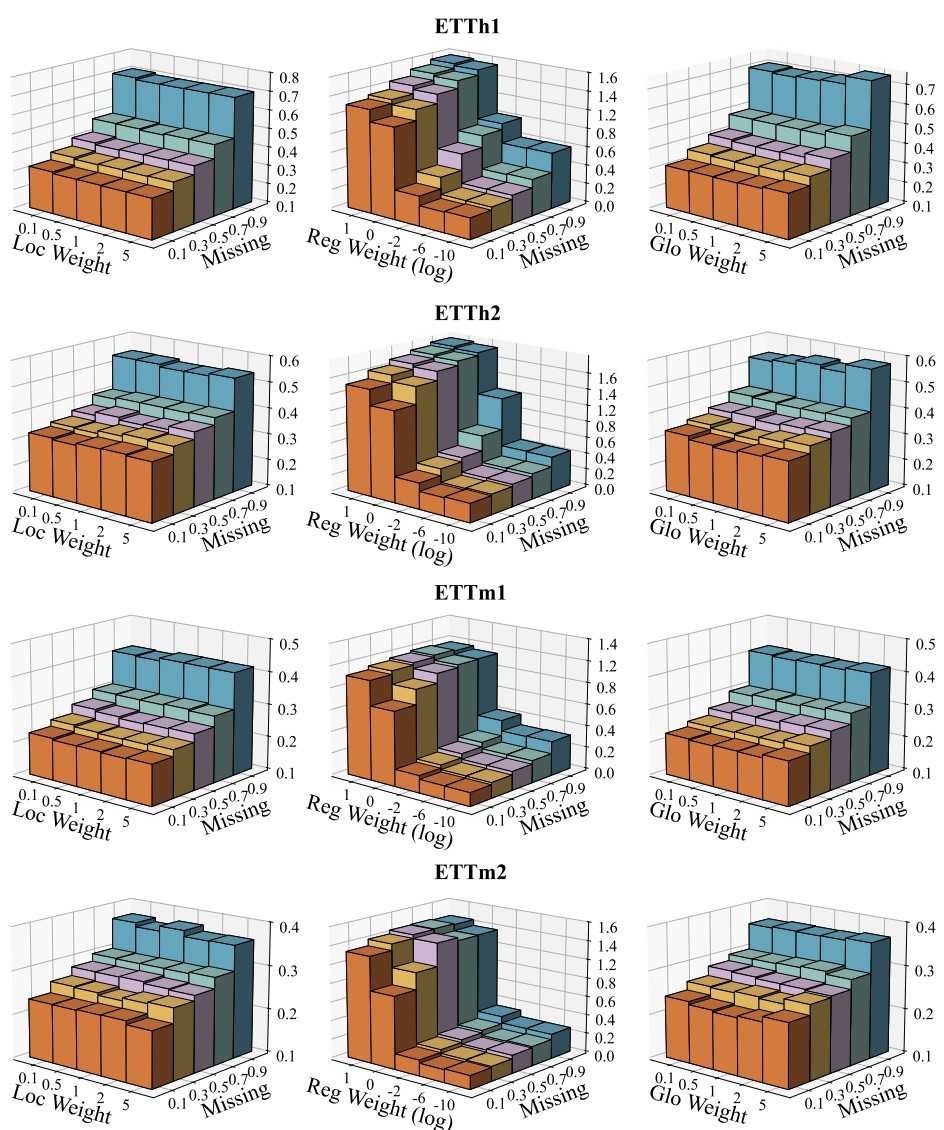

Figure 14: Full sensitivity results on ETTh1, ETTh2, ETTm1, and ETTm2.

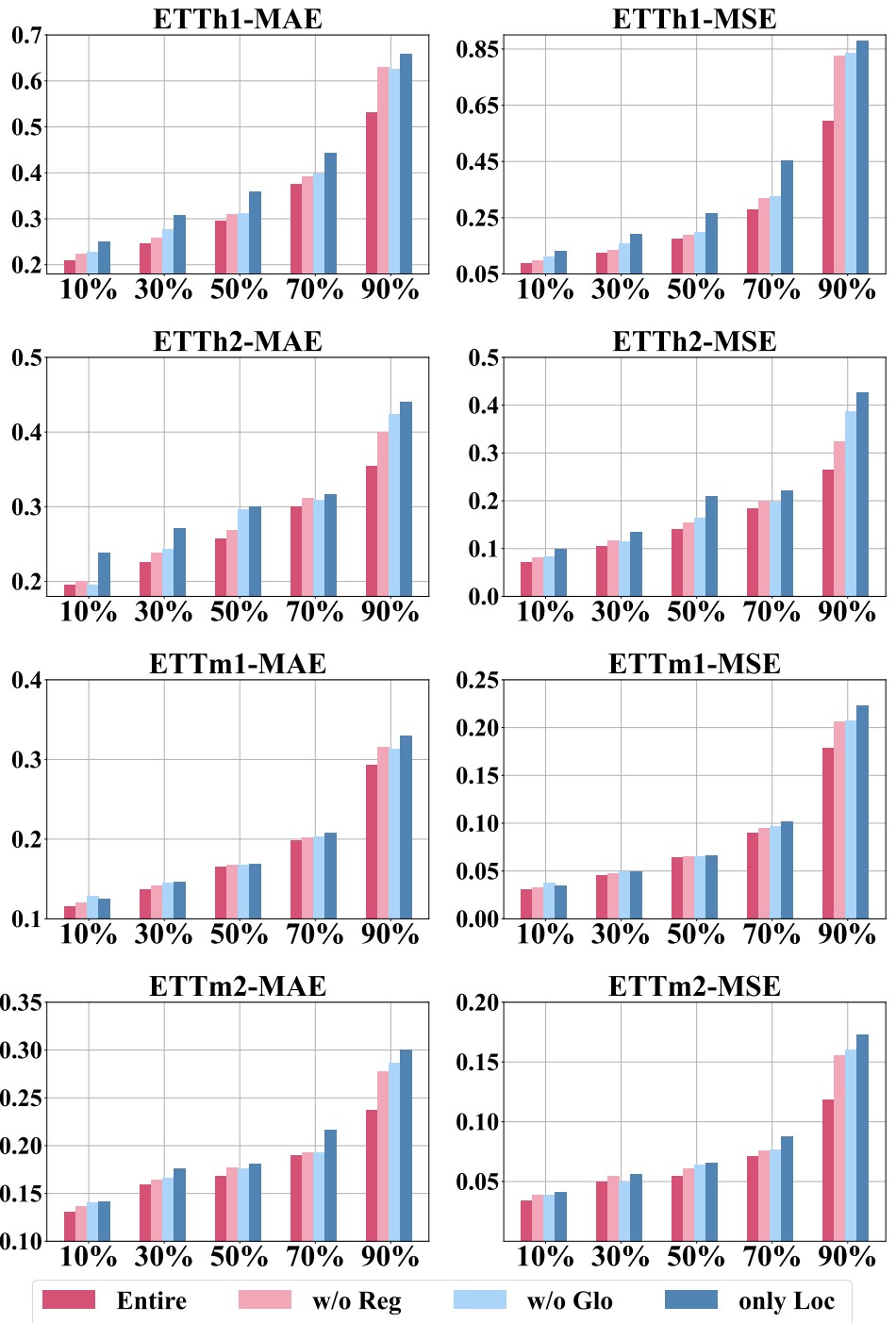

Figure 15: Full ablation study results on ETTh1, ETTh2, ETTm1, and ETTm2.

