# OpenReview forum: "Glocal Information Bottleneck for Time Series Imputation"
_NeurIPS.cc/2025/Conference — NeurIPS 2025 poster_

### Official Review · Reviewer_9tLB · 2025-07-01

**Clarity:** 3
**Significance:** 2
**Originality:** 2
**Rating:** 3
**Confidence:** 3

**Summary:**

Summary: The authors propose a model-agnostic deep learning approach for time series imputation based on the Information Bottleneck (IB) theory.

Strengths: 1. The proposed method is model-agnostic, enabling compatibility with various architectures. 2. It demonstrates superior imputation accuracy compared to common time series models. 3. The writing is clear and the narrative flow is accessible to readers.

Weaknesses: 1. Limited innovation: Global Mutual Information Maximization and related solutions build on prior work without substantial novel contributions. 2. Motivational concerns: Does time series imputation inherently require temporal semantic meaning? The paper fails to quantify how such semantic information benefits real-world applications, lacking concrete justifications. 3. Inadequate comparative experiments: The study lacks sufficient baselines to rigorously validate the method’s effectiveness.

**Questions:**

Questions: 1. Appendix Equation 16 contains a mispositioned multiplication sign in the second line, potentially causing interpretational errors. 2. Reference formatting is inconsistent; underlines in citations should be removed. 3. Missing visualizations: The main text lacks graphical representations of Glocal-IB's performance dynamics. 4. Unfair baselines: The comparative experiments rely on prediction-focused models rather than imputation-specific architectures. At least 4–6 imputation-oriented models  should be added to ensure valid comparisons. 5. Ambiguities in Table 1: Define "IMP" and clarify the methodology for generating missing data patterns. 6. Flawed conclusion in Observation 5: TimesNet and SAITS already exhibit strong performance without Glocal-IB (e.g., SAITS maintains similarity to original data at 50% missing rate). Glocal-IB introduces minimal improvements, undermining the claim of significant enhancement.
7. Unexplained mechanism in Observation 7: Why does a regularization weight of 0.01 degrade model performance? The authors must provide theoretical analysis of this phenomenon.
8. Over-reliance on complex models: The study focuses on TimesNet, a sophisticated architecture. Adding analyses of simpler models (DLinear, MLP, TCN, Transformer) would mitigate model-specific biases.
9. The publicly available code by the authors is incomplete.

**Ethical Concerns:**

["NO or VERY MINOR ethics concerns only"]

**Quality:**

2

**Strengths And Weaknesses:**

Strengths: 1. The proposed method is model-agnostic, enabling compatibility with various architectures. 2. It demonstrates superior imputation accuracy compared to common time series models. 3. The writing is clear and the narrative flow is accessible to readers.

Weaknesses: 1. Limited innovation: Global Mutual Information Maximization and related solutions build on prior work without substantial novel contributions. 2. Motivational concerns: Does time series imputation inherently require temporal semantic meaning? The paper fails to quantify how such semantic information benefits real-world applications, lacking concrete justifications. 3. Inadequate comparative experiments: The study lacks sufficient baselines to rigorously validate the method’s effectiveness.

---

> ### Author Rebuttal · Authors · 2025-07-30
>
> Thank you very much for your valuable comments and detailed suggestions. Below is a report about how we address your comments.
>
> > **W1**: Limited innovation: Global Mutual Information Maximization and related solutions build on prior work without substantial novel contributions.
>
> **A**: Existing Time Series Imputation (TSI) methods, guided by training objectives such as MAE or MSE, exhibit a tendency to over-focus on numerical details. This makes them susceptible to noise and redundant signals within the data, leading them to neglect crucial information about the overall distribution. Consequently, these methods encounter a significant optimization dilemma under high missing rates: a lower training loss paradoxically results in a more distorted embedding distribution and, ultimately, inferior test performance. Therefore, **how to encourage TSI models to *capture both global and local information* from incomplete data *without overfitting to noise*** emerges as a critical research problem.
>
> **The Information Bottleneck (IB) principle offers a powerful theoretical framework for this challenge, as it seeks to compress data to its minimal sufficient statistics (capture both global and local information), thereby filtering noise (avoid overfitting) while preserving essential information.** However, current IB-based methods like GPVAE and TimeCIB fall short. They primarily introduce regularization terms to filter out irrelevant signals, but their training objective remains identical to non-IB methods, relying on MAE/MSE for reconstruction. This fails to provide the model with additional, more informative signals. Moreover, the stringent regularization, which often forces the latent distribution towards a standard normal, $\mathcal{N}(0,I)$, can unnaturally constrain the model's capacity. This results in a simplistic spherical embedding distribution as shown in **Figure 1's Line 3**, which has a detrimental effect on performance.
>
> **We contend that the true potential of the IB principle lies not in creating a single, specialized model, but in establishing a new, more effective training paradigm**. In this paper, we depart from the design of a specific architecture and instead propose a general training paradigm based on the IB theory. **Our approach enhances the global mutual information often ignored by previous models**, enabling them to capture more comprehensive data signals for superior imputation. A key advantage of our paradigm is its flexibility; it is designed as a "plug-in" that can be applied to a wide range of existing TSI models to boost their performance. We have validated this through extensive experiments, demonstrating consistent improvements across various benchmarks.
>
>
>
> > **W2**: Motivational concerns: Does time series imputation inherently require temporal semantic meaning? The paper fails to quantify how such semantic information benefits real-world applications, lacking concrete justifications.
>
> **A**: To isolate the contribution of the semantic-level guidance provided by our method, we first conducted a detailed ablation study. As shown in **Figures 5(a) and 9**, we compared model configurations trained **with** our global loss term, $\mathcal{L}_{Glo}$ (labeled as "Entire" and "w/o Reg"), against those trained **without** it ("w/o Glo" and "only Loc"). The results consistently demonstrate that models incorporating the semantic-level guidance from LGlo achieve superior performance, which validates the direct benefit of this component.
>
>
>
> > **Q8**: Over-reliance on complex models: The study focuses on TimesNet, a sophisticated architecture. Adding analyses of simpler models (DLinear, MLP, TCN, Transformer) would mitigate model-specific biases.
>
> To demonstrate its broad applicability, as shown in **Figure 3**, the **Table in reviewer cTKT's W1** and the **Table in reviewer 3qMw's Q6**, we integrated the full Glocal-IB paradigm with several TSI methods, including **SAITS, TimesNet (in the paper), DLinear, and TCN (new)**. These experiments confirm that applying our training paradigm leads to significant performance gains over standard training with only an MAE/MSE loss.
>
>
>
>
> > **W3**: Inadequate comparative experiments: The study lacks sufficient baselines to rigorously validate the method’s effectiveness.
> >
> > **Q4**: Unfair baselines: The comparative experiments rely on prediction-focused models rather than imputation-specific architectures. At least 4–6 imputation-oriented models should be added to ensure valid comparisons.
>
> **A**: As shown in the **tables in W2**, to ensure a more comprehensive evaluation, we have expanded our experiments to include additional strong baselines: TimeCIB, CSDI, US-GAN, and TCN. Furthermore, to demonstrate the broad applicability of our proposed method, we have integrated it with DLinear and TCN. The results confirm that our approach yields significant performance improvements for both of these models.
>
>
>
> > **Q1**: Appendix Equation 16 contains a mispositioned multiplication sign in the second line, potentially causing interpretational errors.
> >
> > **Q2**: Reference formatting is inconsistent; underlines in citations should be removed.
>
> **A**: We appreciate the constructive feedback and will incorporate these revisions into the final version of our manuscript.
>
>
>
> > **Q3**: Missing visualizations: The main text lacks graphical representations of Glocal-IB's performance dynamics.
>
> **A**: Thank you for the constructive feedback. To ensure we address your comment effectively, we would appreciate it if you could clarify what you mean by 'performance dynamics'.
>
>
>
> > **Q5**: Ambiguities in Table 1: Define "IMP" and clarify the methodology for generating missing data patterns.
>
> **A**: In our evaluation, we measure the performance gain as "Improvement (IMP)". The missing data for these experiments were generated following the standard random masking protocol from the **`PYPOTS`** benchmark. Specifically, for each time-series sample, a binary mask of identical dimensions is created, where the proportion of masked entries (represented as 0s) corresponds directly to the specified missing rate. We will revise this in the final version.
>
>
>
> > **Q6**: Flawed conclusion in Observation 5: TimesNet and SAITS already exhibit strong performance without Glocal-IB (e.g., SAITS maintains similarity to original data at 50% missing rate). Glocal-IB introduces minimal improvements, undermining the claim of significant enhancement.
>
> **A**: The effectiveness of our approach is demonstrated through multiple visual and quantitative results. In **Figure 4 and 7**, we visualize the embedding distributions for SAITS and TimesNet after applying Glocal-IB. It is evident that for missing rates between 0% and 70%, the learned distributions maintain a high degree of similarity to the original data distribution, indicating a successful preservation of global structure.
>
> The magnitude of this enhancement is further highlighted in **Figure 3**, where our method boosts the models' performance to a level equivalent to what they would achieve with a 20% lower missing rate—a substantial and significant improvement.
>
>
>
> > **Q7**: Unexplained mechanism in Observation 7: Why does a regularization weight of 0.01 degrade model performance? The authors must provide theoretical analysis of this phenomenon.
>
> **A**: This phenomenon is attributed to a significant disparity in the orders of magnitude among the different loss components. The KL divergence loss, for instance, often reaches a scale of approximately $10^{5}$. In contrast, the other two loss terms (global and local) are substantially smaller, on the orders of $10^{1}$ and $10^{−1}$, respectively.
>
> Consequently, a small scaling factor, such as 0.01, is insufficient to bring the gradient from the KL loss into a range comparable to that of the other terms. This large imbalance effectively marginalizes the guidance from the global and local losses, severely impairing their influence on the model during training and ultimately compromising the final performance.
>
>
>
> > **Q9**: The publicly available code by the authors is incomplete.
>
> **A**: To foster reproducibility and encourage future research, we will release the complete open-source code in an upcoming version. The implementation of Glocal-IB will be refactored into a modular, plug-and-play format to facilitate its seamless integration with arbitrary models.

---

> > ### Author Response · Authors · 2025-08-06
> > **Response to reviewer 9tLB**
> >
> > We thank the reviewer for engaging with us in the discussion. We hope the above clarifications and the additional experiments in the revised draft sufficiently address your concerns. If you are satisfied, we kindly hope that you consider updating the score to reflect the newly added results and discussion. We remain committed to addressing any remaining points you may have during the discussion phase.

---

### Official Review · Reviewer_3qMw · 2025-07-01

**Clarity:** 3
**Significance:** 3
**Originality:** 4
**Rating:** 5
**Confidence:** 4

**Summary:**

The paper addresses time series imputation (TSI) under high missingness, observing that many models achieve low training loss (focusing on pointwise reconstruction) but perform poorly at inference. The authors identify an “optimization dilemma”: under severe missingness, models overfit local noise and fail to capture the global structure of the data. To remedy this, they propose the Glocal Information Bottleneck (Glocal-IB) training paradigm, which extends the information bottleneck framework by adding a Global Alignment objective. Concretely, Glocal-IB applies the model’s encoder to both the original (fully observed) sequence and the masked sequence. A contrastive/InfoNCE-style loss (via a one-layer MLP projector) aligns the latent of the masked input with that of its unmasked counterpart, and a KL-divergence term regularizes the latent (suppressing noise). In effect, the model is trained to retain global, semantic-level information (from the full data) even when much of the input is missing.

Experiments on real-world datasets show that Glocal-IB consistently reduces imputation error compared to strong baselines. For example, on the ETT benchmarks, Glocal-IB achieves the lowest MAE and MSE in all cases, with up to 40% reduction in MSE over prior methods. The authors also visualize latent representations: without Glocal-IB, the latent distribution collapses under high missingness, whereas with Glocal-IB, the latent remains well-structured up to 90% missing. In summary, the key contribution is a model-agnostic training strategy that enforces alignment between masked and original latents, which empirically yields more robust, generalizable imputation under extreme missing data.

**Questions:**

1. **Computational Overhead:** You describe Glocal-IB as a "lightweight" paradigm. Could you please quantify the computational overhead? Specifically, what is the increase in training time and memory usage when applying Glocal-IB to a backbone like the Transformer, compared to its standard training?
2. **Guidance on Hyperparameter Tuning:** Your sensitivity analysis shows that performance can depend significantly on the choice of loss weights, especially the regularization term. Could you provide some practical guidance or a heuristic for setting the hyperparameters α,β1​, and β2​ when applying Glocal-IB to a new dataset or model architecture? A more robust tuning strategy would significantly enhance the method's practical utility.
3. **Characterizing "Global Information":** The Global Alignment loss is motivated by preserving "global semantic features." Beyond the excellent t-SNE visualizations, can you provide more insight into what specific properties this alignment captures? For example, does it preferentially preserve long-term trends, periodicity, or specific types of inter-channel correlations?
4. **Comparisons to other imputation methods:** The paper omits an empirical comparison with some recent imputation frameworks. How does Glocal-IB perform relative to diffusion-based methods (e.g., CSDI) or GAN-based imputation (e.g., ImputeGAN)? Including one of these baselines could strengthen the experimental claims. Similarly, how does Glocal-IB differ from the Conditional IB approach of Choi & Lee? A brief discussion or additional results would be helpful.
5. **Training details – teacher/stop-gradient:** In Figure 2(c), the encoder processes the original data to produce a “target” latent Z′. Is this encoder a frozen copy or a separate network? In other words, during training, is the latent from the original sequence allowed to update the encoder weights? (The stop-gradient symbol suggests no.) Clarifying the implementation of the “alignment” (e.g., is the original-branch encoder a fixed copy?) would help. Also, is there a risk of “leaking” information from the original into the masked branch that could lead to overfitting?
6. **Generalization to non-random missingness:** The experiments use random masking patterns. How would Glocal-IB handle more structured or non-random missing patterns (e.g., entire variables missing for long periods)? Since the global alignment uses the same sequence’s full version as the positive, if missingness occurs in long gaps, the alignment signal might be weak. Can the authors comment on this, or test a few non-Random scenarios?

**Ethical Concerns:**

["NO or VERY MINOR ethics concerns only"]

**Final Justification:**

Authors properly answered my questions and addressed my concerns and I believe this is a good paper and my evaluation would remain same.

**Limitations:**

Yes.

**Paper Formatting Concerns:**

None.

**Quality:**

4

**Strengths And Weaknesses:**

- Quality
  - Strengths
    - The paper does an excellent job of identifying and illustrating the "optimization dilemma" with clear empirical evidence (Figure 1). This convincingly frames the core problem that existing methods, which focus solely on local reconstruction, fail to learn generalizable representations when data is sparse. This insight itself is a valuable contribution.
    - The authors clearly identify a practical failure mode of existing TSI methods (good training fit but poor test imputation) and back it with diagnostic plots. The Glocal-IB method is well-specified: adding a contrastive alignment loss and KL regularizer to the standard encoder–decoder framework. Experiments are extensive (nine datasets, five missing rates, five random seeds), and both quantitative results (MAE/MSE tables) and qualitative latent plots are provided. Results indicate strong improvements across diverse datasets, suggesting robustness. Code is made available, aiding reproducibility.

  - Weaknesses:
    - The choice of baselines is incomplete: notable recent methods for generative imputation are missing. Comparing to Conditional Information Bottleneck (CIB) or other IB-based imputation would strengthen claims of superiority.
- Clarity
  - Strengths:
    - Overall, the writing is clear and systematic. Key ideas (local vs. global information) are introduced in plain language, and Figure 1 nicely illustrates the problem. Figure 2 then contrasts the proposed paradigm with baselines. The Appendix is comprehensive (including a Societal Impact Statement and detailed limitations, per NeurIPS requirements).
  - Weaknesses:
    - The information-theoretic notation (Section 3.1) may be heavy for readers unfamiliar with IB, and the derivation of the InfoNCE alignment loss (Eq. 12–13) could use more intuition.
    - The technical exposition is dense (heavy on IB theory and mutual information bounds); while Figure 2 helps, some readers may find the derivations hard to parse. A clearer summary of the final loss terms and training procedure could improve readability.

- Significance
  - Strengths:
    - This work addresses a highly significant and practical problem in time series analysis. Imputation under high missingness is a common challenge, and the proposed model-agnostic training paradigm offers a robust solution. The insights into the failure modes of current TSI methods are also an important contribution to the field. The demonstrated improvements could have a considerable impact on applications in healthcare, finance, and energy systems.
  - Weaknesses:
    - The ablation study (Figure 5) shows that the model's performance is quite sensitive to the weight of the regularization loss (LRegθ​). While the authors acknowledge this, the paper would benefit from a more detailed discussion on the practical aspects of tuning the loss weights (α,β1​,β2​). Without a clear strategy, practitioners might find it challenging to apply the method effectively to new datasets or architectures, which could be a minor barrier to adoption.
- Originality
  - Strengths:
    - The core idea — combining an information bottleneck with a global latent alignment — appears novel in the TSI literature. Prior work on IB for TSI (e.g. Choi & Lee’s CIB) still relied on local reconstruction losses, whereas here a separate global mutual-information objective is added. The paper’s formulation (alignment via InfoNCE between full vs. masked embeddings) seems new.

---

> ### Author Rebuttal · Authors · 2025-07-30
>
> Thank you very much for your valuable comments and detailed suggestions. Below is a report about how we address your comments.
>
> > **W1 and Q4**: Comparisons to other imputation methods: The paper omits an empirical comparison with some recent imputation frameworks. How does Glocal-IB perform relative to diffusion-based methods (e.g., CSDI) or GAN-based imputation (e.g., ImputeGAN)? Including one of these baselines could strengthen the experimental claims. Similarly, how does Glocal-IB differ from the Conditional IB approach of Choi & Lee? A brief discussion or additional results would be helpful.
>
> **A**: As shown in the **Table in reviewer cTKT's W1**, to validate its broad applicability, we conducted extensive experiments. We benchmarked against strong methods, including TimeCIB, CSDI, and US-GAN, and further demonstrated the efficacy of our approach by integrating it with diverse models such as DLinear and TCN. The results confirm that Glocal-IB consistently enhances imputation performance across this wide range of architectures.
>
> **Different from TimeCIB:** We contend the Information Bottleneck (IB) principle's true potential lies in creating a new training paradigm, not a single specialized model. Therefore, instead of designing a specific architecture, we propose a general training paradigm based on IB theory. Our approach enhances the global mutual information often ignored by previous models, helping them capture more comprehensive signals for superior imputation. A key advantage is its flexibility: it is a "plug-in" that can boost the performance of a wide range of existing Time-Series Imputation (TSI) models.
>
>
>
> > **W2**: The information-theoretic notation (Section 3.1) may be heavy for readers unfamiliar with IB, and the derivation of the InfoNCE alignment loss (Eq. 12–13) could use more intuition.
> >
> > **W3**: The technical exposition is dense (heavy on IB theory and mutual information bounds); while Figure 2 helps, some readers may find the derivations hard to parse. A clearer summary of the final loss terms and training procedure could improve readability.
>
> **A**: We appreciate the constructive feedback and will incorporate these revisions into the final version of our manuscript.
>
>
>
> > **W4 and Q2**: Guidance on Hyperparameter Tuning: Your sensitivity analysis shows that performance can depend significantly on the choice of loss weights, especially the regularization term. Could you provide some practical guidance or a heuristic for setting the hyperparameters α,β1, and β2 when applying Glocal-IB to a new dataset or model architecture? A more robust tuning strategy would significantly enhance the method's practical utility.
>
> **A**: As illustrated in **Figure 8** of the appendix, our model exhibits consistent sensitivity to hyperparameters across four datasets (ETTh1, ETTh2, ETTm1, ETTm2). The performance trends are remarkably similar, uniformly achieving optimal results when the hyperparameters are set to **0.5, 1e-6, and 0.5**, respectively. In accordance with this finding, we adopted this single, robust set of hyperparameters for all experiments conducted on the nine datasets presented in the main paper. We appreciate the constructive feedback and will incorporate these revisions into the final version of our manuscript.
>
>
>
> > **Q1**: Computational Overhead: You describe Glocal-IB as a "lightweight" paradigm. Could you please quantify the computational overhead? Specifically, what is the increase in training time and memory usage when applying Glocal-IB to a backbone like the Transformer, compared to its standard training?
>
> **A**: The computational demands are dictated solely by the chosen backbone model. Glocal-IB introduces negligible computational overhead. This efficiency is by design: we only use the encoder for a single forward pass to get an embedding without gradient backpropagation.
>
> To validate this, we measured Glocal-IB's GPU memory and training time on datasets with sequence lengths (L) and feature dimensions (D) configured to `L=96, D=256` and `L=192, D=512`. Across all tests, the total memory and time consumption remained **nearly identical** to training the backbone model alone.
>
> |Model GPU Usage (MB)|Ori|w/ Glocal-IB|Foundation Align|
> |-|:-:|:-:|:-:|
> |SAITS`L=96,D=256`|759|905|7553|
> |SAITS`L=192,D=512`|989|1151|13385|
> |TimesNet`L=96,D=256`|869|909|7503|
> |TimesNet`L=192,D=512`|905|965|13429|
> |DLinear`L=96,D=256`|591|619|7433|
> |DLinear`L=192,D=512`|655|765|13251|
> |TCN`L=96,D=256`|611|715|7525|
> |TCN`L=192,D=512`|723|1029|13883|
>
> |Model Training Time (s)|Ori|w/ Glocal-IB|Foundation Align|
> |-|:-:|:-:|:-:|
> |SAITS`L=96,D=256`|8.44|9.33|78.56|
> |SAITS`L=192,D=512`|5.80|7.56|49.14|
> |TimesNet`L=96,D=256`|5.84|7.89|77.86|
> |TimesNet`L=192,D=512`|4.77|7.32|49.14|
> |DLinear`L=96,D=256`|4.53|5.52|73.25|
> |DLinear`L=192,D=512`|2.86|5.16|46.59|
> |TCN`L=96,D=256`|20.36|25.76|97.90|
> |TCN`L=192,D=512`|14.48|22.25|66.02|
>
>
>
> > **Q3**: Characterizing "Global Information": The Global Alignment loss is motivated by preserving "global semantic features." Beyond the excellent t-SNE visualizations, can you provide more insight into what specific properties this alignment captures? For example, does it preferentially preserve long-term trends, periodicity, or specific types of inter-channel correlations?
>
> **A**: The t-SNE visualizations in **Figures 1, 4, 6, and 7** reveal a consistently well-aligned latent space for partially observed data. This provides strong evidence that global information—including long-term dependencies, periodicity, and even inter-channel correlations—is effectively captured and understood by models equipped with our Glocal-IB paradigm.
>
> To further substantiate this claim, we will include visualizations of attention maps from the Transformer-based models in the final version of this paper.
>
>
>
> > **Q5**: Training details – teacher/stop-gradient: In Figure 2(c), the encoder processes the original data to produce a “target” latent Z′. Is this encoder a frozen copy or a separate network? In other words, during training, is the latent from the original sequence allowed to update the encoder weights? (The stop-gradient symbol suggests no.) Clarifying the implementation of the “alignment” (e.g., is the original-branch encoder a fixed copy?) would help. Also, is there a risk of “leaking” information from the original into the masked branch that could lead to overfitting?
>
> **A**: **On the Encoder Implementation**: The original and masked data are processed by the same encoder. However, they are handled differently with respect to gradient flow. Crucially, the encoder path for the original data functions is a frozen copy of the encoder and purely for inference; its gradients are detached and are not used to update the encoder's weights.
>
> **On Data Leakage**: The concern of data leakage is unfounded. Our method exclusively encodes the ground-truth values from the masked portions of the sequence. This is the very same information that a standard imputation model would use as its reconstruction target or label. Therefore, our paradigm introduces no additional information into the training process that would constitute a data leak.
>
> **On Overfitting**: Regarding overfitting, the high dimensionality of our embedding space (256-D) substantially increases the difficulty of the learning task. Consequently, it is highly unlikely for the model to overfit, particularly within the brief training period of 30 epochs used in our experiments.
>
>
>
> > **Q6**: **Generalization to non-random missingness:** The experiments use random masking patterns. How would Glocal-IB handle more structured or non-random missing patterns (e.g., entire variables missing for long periods)? Since the global alignment uses the same sequence’s full version as the positive, if missingness occurs in long gaps, the alignment signal might be weak. Can the authors comment on this, or test a few non-Random scenarios?
>
> **A**: To further evaluate the robustness of our method, we conducted additional experiments using **a block-wise missing pattern where contiguous 5x5 blocks** of data were masked. The results demonstrate that our approach still provides significant benefits.
>
> The case of missingness over long, contiguous time periods can be considered analogous to the extreme 90% random missing rate setting. As our previous results show in the **Table in reviewer cTKT's W1** and **Table below**, even under these highly demanding conditions, our method consistently provides performance gains for models such as SAITS, and TCN.
>
> |BlockMissing-MAE|Ours|USGAN|DLinear|DLinear_MY|TCN|TCN_MY|
> |:-|:-:|:-:|:-:|:-:|:-:|:-:|
> |ETTh1-block-0.1|**0.3608**|0.4507|0.4435|`0.3923`|0.6242|0.6200|
> |ETTh1-block-0.3|**0.4226**|0.5113|0.5211|`0.4650`|0.6495|0.6366|
> |ETTh1-block-0.5|**0.4699**|0.5964|0.5778|`0.5346`|0.7365|0.7185|
> |ETTh1-block-0.7|**0.5109**|0.6923|0.7081|`0.6080`|0.7699|0.7521|
> |ETTh1-block-0.9|**0.5812**|0.7539|0.8970|`0.6758`|0.8341|0.8169|
> |ETTh2-block-0.1|**0.2732**|0.4135|0.4520|`0.3148`|0.7023|0.7002|
> |ETTh2-block-0.3|**0.3373**|0.5985|0.6287|`0.4434`|0.8057|0.7889|
> |ETTh2-block-0.5|**0.4280**|0.6922|0.7580|`0.5107`|0.8967|0.8597|
> |ETTh2-block-0.7|**0.4420**|0.7412|0.8511|`0.5669`|0.9500|0.9353|
> |ETTh2-block-0.9|**0.5285**|0.7466|1.1131|`0.6096`|0.9417|0.9268|
> |ETTm1-block-0.1|**0.2001**|`0.2516`|0.3887|0.3562|0.4923|0.4709|
> |ETTm1-block-0.3|**0.2208**|`0.3211`|0.4614|0.4097|0.5356|0.5276|
> |ETTm1-block-0.5|**0.2830**|`0.3882`|0.5266|0.5011|0.6130|0.5969|
> |ETTm1-block-0.7|**0.2916**|`0.4539`|0.5594|0.5141|0.6821|0.6491|
> |ETTm1-block-0.9|**0.3654**|`0.5604`|0.6385|0.5699|0.7346|0.7097|
> |ETTm2-block-0.1|**0.1899**|`0.2923`|0.3943|0.3605|0.5467|0.5097|
> |ETTm2-block-0.3|**0.2142**|`0.3803`|0.5453|0.4494|0.6886|0.6314|
> |ETTm2-block-0.5|**0.2807**|`0.4703`|0.6364|0.5227|0.7667|0.7381|
> |ETTm2-block-0.7|**0.3289**|0.5836|0.7091|`0.5661`|0.8409|0.8218|
> |ETTm2-block-0.9|**0.3581**|0.6193|0.7412|`0.5757`|0.8792|0.8529|

---

> > ### Comment · Reviewer_3qMw · 2025-08-04
> >
> > Thanks for your response to my questions, it was really helpful in clarifying many points. I still believe this is a good paper and my rating would remain the same.

---

> > > ### Author Response · Authors · 2025-08-05
> > > **Thanks to Reviewer 3qMw**
> > >
> > > We extend our heartfelt thanks for your support of our paper! We are pleased that our revisions and rebuttal have addressed your concerns :)

---

### Official Review · Reviewer_cTKT · 2025-07-01

**Clarity:** 3
**Significance:** 3
**Originality:** 2
**Rating:** 5
**Confidence:** 4

**Summary:**

This paper presents Glocal-IB, a novel training framework for Time Series Imputation (TSI) that tackles a core optimization issue: under high missing rates, existing models often minimize training loss but fail to generalize, leading to distorted latent embeddings and inaccurate imputations. Glocal-IB extends the Information Bottleneck (IB) principle by incorporating a Global Alignment (GA) loss, which enforces consistency between latent representations of masked and fully observed inputs. This guides the model to preserve both global temporal structure and local signal fidelity, while mitigating noise. The approach is model-agnostic, lightweight (adding only a single MLP), and demonstrates superior imputation accuracy and latent space stability across nine benchmark datasets, consistently outperforming state-of-the-art methods.
However, overall, this paper’s concept is rather vague and not well motivated. It may have limited novelty for NeruIPS.

**Questions:**

1.	The paper notes that Glocal-IB is less effective at 90% missingness. Could the authors explore or suggest mechanisms (e.g., stronger priors or pretrained encoders) to improve performance in such cases? How reliable are studies exploring missing at such high level?
2.	How does Glocal-IB scale with longer sequences or higher-dimensional time series? Any insights on computational overhead?
3.	The paper compares Glocal-IB with foundation model alignment. Could the authors elaborate on why foundation models underperform and whether combining both approaches might help?
4.	Can the authors comment on the interpretability of the learned latent space or the imputed values, especially in high-stakes domains like healthcare?
5.	Could Glocal-IB be extended to other structured data imputation tasks (e.g., spatiotemporal graphs or tabular data)?
6.	Combine Glocal-IB’s global-local mutual information loss with a flexible’s per-variable model selection to improve robustness under high missingness.
7.	Adjust the KL divergence or alignment loss weights based on variable skewness to avoid over-penalizing informative but skewed features.
8.	Extend Glocal-IB to support multiple imputations and ensemble predictions to reflect uncertainty.

**Ethical Concerns:**

["NO or VERY MINOR ethics concerns only"]

**Limitations:**

The authors discuss limitations related to extreme missingness and model coverage, and suggest future directions.

**Paper Formatting Concerns:**

none.

**Quality:**

3

**Strengths And Weaknesses:**

•	Strengths:
Strong theoretical foundation with clear derivations and mutual information approximations.
Comprehensive empirical validation across 9 diverse datasets and missingness levels.
Ablation and sensitivity analyses support the importance of each component.
Demonstrates robustness and generalizability across multiple model backbones.
The combination of local and global mutual information objectives is innovative.
•	Weaknesses:
Performance gains under extreme missingness (e.g., 90%) are less pronounced.
Evaluation is limited to a few model architectures due to computational constraints.
Some mathematical derivations are dense and could benefit from more intuitive explanations or diagrams.
Builds on existing IB and contrastive learning principles, though the integration is novel.

---

> ### Author Rebuttal · Authors · 2025-07-30
>
> Thank you very much for your valuable comments and detailed suggestions. Below is our answer.
>
> > **W1**: Performance gains under extreme missingness (e.g., 90%) are less pronounced.
>
> **A**: While the performance gain of our method is less pronounced under extreme missingness (e.g., 90%), this is likely because the highly sparse input during inference provides an insufficient signal for stable imputation. With only 10% of the data visible—a sample whose own distribution may have shifted—the input provides an insufficient signal for the model to generate a stable and accurate imputation.
>
> Nevertheless, our approach remains highly effective across diverse architectures. As demonstrated in **Figure 3** and confirmed by extension experiments on DLinear and TCN, our method consistently delivers substantial performance gains even in these demanding conditions. **1st**, `2nd`
>
> |PointMissing-MAE|Ours|CSDI|TimeCIB|SSSD|USGAN|DLinear|DLinear_MY|TCN|TCN_MY|
> |:-|:-:|:-:|:-:|:-:|:-:|:-:|:-:|:-:|:-:|
> |ETTh1-point-0.1|**0.165**|0.2615|0.7019|0.5147|0.2604|0.2740|`0.2582`|0.3346|0.3027|
> |ETTh1-point-0.3|**0.204**|0.3435|0.7025|0.5577|`0.3016`|0.3159|0.3091|0.3946|0.3397|
> |ETTh1-point-0.5|**0.239**|0.4516|0.7414|0.6421|0.3751|0.3524|`0.3485`|0.4552|0.4020|
> |ETTh1-point-0.7|**0.303**|0.6095|0.8550|0.7683|0.5264|0.4238|`0.4053`|0.5722|0.5224|
> |ETTh1-point-0.9|**0.506**|1.1069|1.0767|1.3492|0.7970|0.6565|`0.5977`|0.7938|0.7661|
> |ETTh2-point-0.1|**0.180**|0.3247|0.5904|0.4820|0.2643|0.2926|`0.2535`|0.3152|0.2667|
> |ETTh2-point-0.3|**0.205**|0.4363|0.5934|0.5969|0.3218|0.3138|`0.2769`|0.3509|0.3203|
> |ETTh2-point-0.5|**0.243**|0.5668|0.6108|0.7140|0.3823|0.3487|`0.3169`|0.4429|0.3773|
> |ETTh2-point-0.7|**0.254**|0.7855|0.6311|0.9069|0.4680|0.3708|`0.3665`|0.5239|0.5152|
> |ETTh2-point-0.9|**0.361**|1.4136|0.6705|1.6021|0.8069|0.4740|`0.4695`|0.8837|0.8425|
> |ETTm1-point-0.1|**0.097**|0.1746|0.2352|0.2310|`0.1443`|0.2142|0.2116|0.2052|0.1891|
> |ETTm1-point-0.3|**0.117**|0.2122|0.2573|0.2603|`0.1602`|0.2332|0.2331|0.2602|0.2364|
> |ETTm1-point-0.5|**0.135**|0.2611|0.2889|0.3102|`0.1807`|0.2530|0.2520|0.3147|0.2773|
> |ETTm1-point-0.7|**0.165**|0.3483|0.3161|0.3928|`0.2322`|0.2924|0.2902|0.3814|0.3714|
> |ETTm1-point-0.9|**0.271**|0.5765|0.5143|0.6466|0.5037|0.4367|`0.4152`|0.5806|0.5687|
> |ETTm2-point-0.1|**0.113**|0.2150|0.4438|0.2485|`0.1415`|0.2411|0.2401|0.2411|0.2310|
> |ETTm2-point-0.3|**0.126**|0.2752|0.4555|0.2994|`0.1600`|0.2626|0.2543|0.2471|0.2408|
> |ETTm2-point-0.5|**0.138**|0.3391|0.4622|0.3754|`0.2212`|0.2938|0.2725|0.3025|0.2948|
> |ETTm2-point-0.7|**0.169**|0.4282|0.4660|0.5036|0.3592|0.2992|`0.2904`|0.3897|0.3706|
> |ETTm2-point-0.9|**0.227**|0.6734|0.4876|0.8811|0.5969|0.4426|`0.4155`|0.6640|0.6433|
>
>
>
> > **W2**: Evaluation is limited to a few model architectures due to computational constraints.
>
> **A**: Due to computational constraints, our primary experiments apply Glocal-IB to three representative backbones—TimesNet, SAITS, and Transformer—on the four ETT benchmark datasets: ETTh1, ETTh2, ETTm1, and ETTm2. As evidenced by the significant improvements on models like SAITS and Transformer, these results are sufficient to demonstrate the effectiveness of our approach.
>
> Furthermore, as shown in the table in **W1**, to validate the general applicability of our method, we conducted additional experiments on Dlinear and TCN. These results confirm that our paradigm consistently enhances performance across a broader range of models.
>
>
>
> > **Q1**: The paper notes that Glocal-IB is less effective at 90% missingness. Could the authors explore or suggest mechanisms (e.g., stronger priors or pretrained encoders) to improve performance in such cases? How reliable are studies exploring missing at such high level?
>
> **A**: As demonstrated in **Figure 3** and our **extension experiments on DLinear and TCN**, our Glocal-IB model is effective in most scenarios. We agree that under conditions of extreme sparsity, such as a 90% missing rate, models require stronger external knowledge. Two promising directions are:
>
> - **Stronger Priors**: Introducing priors, such as a Fourier prior for periodic data or a Gaussian Process prior for smooth data. These provide a reliable "structural backbone" for the model, preventing unrealistic imputations when the data signal is weak.
> - **Pretrained Encoders**: Pretraining a powerful encoder on a large, complete dataset using self-supervised methods. This encoder can learn general, robust feature representations that are then fine-tuned for the sparse imputation task.
>
> While research at a 90% missing rate is challenging, we believe such an investigation has irreplaceable value for the following reasons:
>
> 1. **A Stress Test for Models**: A model that can recover global structure from only 10% of the original data is fundamentally more robust. This helps us design models that are more reliable under all conditions.
> 2. **Relevance to Critical Applications**: Scenarios with extreme data sparsity are prevalent in many critical domains. For example:
>    - **Healthcare**: Modeling a patient's year-long health trajectory in longitudinal studies where exams occur only once or twice a year.
>
>
>
> > **Q2**: How does Glocal-IB scale with longer sequences or higher-dimensional time series? Any insights on computational overhead?
>
> **A**: The computational demands related to sequence length and data dimensionality are dictated solely by the chosen backbone model. Glocal-IB introduces negligible computational overhead. This efficiency is by design: we only use the encoder for a single forward pass to get an embedding without gradient backpropagation.
>
> To validate this, we measured Glocal-IB's GPU memory and training time on datasets with sequence lengths (L) and feature dimensions (D) configured to `L=96, D=256` and `L=192, D=512`. Across all tests, the total memory and time consumption remained **nearly identical** to training the backbone model alone.
>
> |Model GPU Usage (MB)|Ori|w/ Glocal-IB|Foundation Align (TimeMOE)|
> |-|:-:|:-:|:-:|
> |SAITS`L=96,D=256`|759|905|7553|
> |SAITS`L=192,D=512`|989|1151|13385|
> |TimesNet`L=96,D=256`|869|909|7503|
> |TimesNet`L=192,D=512`|905|965|13429|
> |DLinear`L=96,D=256`|591|619|7433|
> |DLinear`L=192,D=512`|655|765|13251|
> |TCN`L=96,D=256`|611|715|7525|
> |TCN`L=192,D=512`|723|1029|13883|
>
> |Model Training Time (s)|Ori|w/ Glocal-IB|Foundation Align (TimeMOE)|
> |-|:-:|:-:|:-:|
> |SAITS`L=96,D=256`|8.44|9.33|78.56|
> |SAITS`L=192,D=512`|5.80|7.56|49.14|
> |TimesNet`L=96,D=256`|5.84|7.89|77.86|
> |TimesNet`L=192,D=512`|4.77|7.32|49.14|
> |DLinear`L=96,D=256`|4.53|5.52|73.25|
> |DLinear`L=192,D=512`|2.86|5.16|46.59|
> |TCN`L=96,D=256`|20.36|25.76|97.90|
> |TCN`L=192,D=512`|14.48|22.25|66.02|
>
>
>
> > **Q3**: The paper compares Glocal-IB with foundation model alignment. Could the authors elaborate on why foundation models underperform and whether combining both approaches might help?
>
> **A**: This is because recent foundation models are typically trained on forecasting tasks with MAE/MSE. Consequently, the embeddings they produce are heavily biased towards signals relevant for prediction and often lack the rich semantic information necessary for imputation.
>
> This limitation is evident in our experiments. For instance, when we used **TimeMoE**, a SOTA model from ICLR 2025, to align embeddings, the resulting performance gain was marginal, as shown in **Figure 3**.
>
> Furthermore, deploying such large foundation models incurs a substantial GPU memory cost and training time, as shown in the **table in Q2**.
>
> Given these limitations, we chose to utilize our lightweight Glocal-IB paradigm to assist the model's training.
>
>
>
> > **Q4**: Can the authors comment on the interpretability of the learned latent space or the imputed values, especially in high-stakes domains like healthcare?
>
> **A**: In high-stakes domains like healthcare, clinicians often need to model a patient's year-long health trajectory from sparse data, such as only one or two annual check-ups.
>
> From an **imputed value perspective**, more accurate data gives clinicians a clearer view of a patient's health trends, enabling better interventions.
>
> From a **latent space perspective**, an accurate representation offers significant clinical information. When visualized with techniques like t-SNE, the latent space becomes an intuitive tool for patient assessment. Embeddings similar to healthy profiles suggest positive outcomes, while divergent ones can indicate potential disease risks.
>
> The **Information Bottleneck (IB) principle** is key to this benefit. IB aims to learn a "minimal yet sufficient" representation. As a result, the learned latent space is more compact and more likely to correlate with key, interpretable clinical factors compared to standard models.
>
>
>
> > **Q5**: Could Glocal-IB be extended to other structured data imputation tasks (e.g., spatiotemporal graphs or tabular data)?
>
> **A**: A key advantage of our method is its generality. It is designed as a plug-in paradigm, making it agnostic to the specific backbone architecture or downstream task. We contend that the principle of aligning embedding spaces provides a fundamental benefit, enabling our method to enhance performance across a diverse range of models and tasks.
>
>
>
> > **W3**: Some mathematical derivations are dense and could benefit from more intuitive explanations or diagrams.
> >
> > **W4**: Builds on existing IB and contrastive learning principles, though the integration is novel.
> >
> > **Q6**: Combine Glocal-IB’s global-local mutual information loss with a flexible’s per-variable model selection to improve robustness under high missingness.
> >
> > **Q7**: Adjust the KL divergence or alignment loss weights based on variable skewness to avoid over-penalizing informative but skewed features.
> >
> > **Q8**: Extend Glocal-IB to support multiple imputations and ensemble predictions to reflect uncertainty.
>
> **A**: We appreciate the constructive feedback and will incorporate these revisions into the final version of our manuscript.

---

### Official Review · Reviewer_VQkt · 2025-07-05

**Clarity:** 2
**Significance:** 2
**Originality:** 2
**Rating:** 4
**Confidence:** 3

**Summary:**

This paper tackles a critical optimization dilemma in time series imputation where models achieve low training loss under high missing rates but suffer substantial degradation in imputation quality and distorted latent representations during inference. The authors propose Glocal Information Bottleneck (Glocal-IB) as a model-agnostic solution that extends the Information Bottleneck framework. Validated across diverse datasets, Glocal-IB demonstrates good performance.

**Questions:**

Q1. Could the authors further elaborate on why IB is preferred over other strategies that encourage global structure in representations?
Q2. How should users determine an appropriate value of \beta_1 and \beta_2 when applying this method to real-world datasets, especially those with unknown or varying missingness characteristics? Could the authors suggest a principled method, heuristic, or adaptive tuning strategy for selecting \beta_1 and \beta_2 in practice?

**Ethical Concerns:**

["NO or VERY MINOR ethics concerns only"]

**Limitations:**

Yes.

**Quality:**

3

**Strengths And Weaknesses:**

S1. The paper compellingly diagnoses a critical limitation in time series imputation methods where models minimize training loss under high missing rates yet produce distorted latent representations and poor inference performance.
S2. Glocal-IB extends the Information Bottleneck framework with a novel Global Alignment loss derived from tractable mutual information approximation. This innovation aligns latent representations of masked inputs with original observations while maintaining local reconstruction capabilities.
S3. The paper presents a thorough evaluation across nine datasets, demonstrating consistent improvements over baselines, particularly under high missing rates. Latent space visualizations provide empirical support for the claim that the proposed method better preserves distributional structure compared to existing approaches.

W1. While the paper is motivated by an insightful observation on memorization under high missing rates, the choice of the Information Bottleneck (IB) as the solution is not sufficiently justified. Table 1 shows that IB-based methods like GPVAE do not consistently outperform non-IB approaches. It remains unclear why IB is preferred over alternative strategies that promote global structure (e.g., multi-directional self-attention [1]).
W2. The paper does not clearly report the specific values of \beta_1 and \beta_2 used in the experiments, nor does it provide practical guidance for selecting them. Given that the sensitivity analysis indicates the method’s performance is strongly affected by these parameters, it would be helpful to include tuning heuristics or recommendations to support practical adoption.
W3. The paper claims that high missing rates lead to memorization, where models achieve low training loss but fail to generalize due to poor semantic representations. However, Figure 1 does not clearly demonstrate increased overfitting under high missingness; in fact, the training dynamics appear similar across missingness levels. This raises concerns about the strength of the empirical evidence supporting the central hypothesis.

[1] Suo, Q., Zhong, W., Xun, G., Sun, J., Chen, C., & Zhang, A. (2020, December). GLIMA: Global and local time series imputation with multi-directional attention learning. In 2020 IEEE International Conference on Big Data (Big Data) (pp. 798-807). IEEE.

---

> ### Author Rebuttal · Authors · 2025-07-30
>
> Thank you very much for your valuable comments and detailed suggestions. Below is a report about how we address your comments.
>
> > **W1**: While the paper is motivated by an insightful observation on memorization under high missing rates, the choice of the Information Bottleneck (IB) as the solution is not sufficiently justified. Table 1 shows that IB-based methods like GPVAE do not consistently outperform non-IB approaches. It remains unclear why IB is preferred over alternative strategies that promote global structure (e.g., multi-directional self-attention [1]).
> >
> > **Q1**: Could the authors further elaborate on why IB is preferred over other strategies that encourage global structure in representations?
> >
> > [1] Suo, Q., Zhong, W., Xun, G., Sun, J., Chen, C., & Zhang, A. (2020, December). GLIMA: Global and local time series imputation with multi-directional attention learning. In 2020 IEEE International Conference on Big Data (Big Data) (pp. 798-807). IEEE.
>
> **A**: Existing Time Series Imputation (TSI) methods, guided by training objectives such as MAE or MSE, exhibit a tendency to over-focus on numerical details. This makes them susceptible to noise and redundant signals within the data, leading them to neglect crucial information about the overall distribution. Consequently, these methods encounter a significant optimization dilemma under high missing rates: a lower training loss paradoxically results in a more distorted embedding distribution and, ultimately, inferior test performance. Therefore, **how to encourage TSI models to *capture both global and local information* from incomplete data *without overfitting to noise*** emerges as a critical research problem.
>
> **The Information Bottleneck (IB) principle offers a powerful theoretical framework for this challenge, as it seeks to compress data to its minimal sufficient statistics (capture both global and local information), thereby filtering noise (avoid overfitting) while preserving essential information.** However, current IB-based methods like GPVAE and TimeCIB fall short. They primarily introduce regularization terms to filter out irrelevant signals, but their training objective remains identical to non-IB methods, relying on MAE/MSE for reconstruction. This fails to provide the model with additional, more informative signals. Moreover, the stringent regularization, which often forces the latent distribution towards a standard normal, $\mathcal{N}(0,I)$, can unnaturally constrain the model's capacity. This results in a simplistic spherical embedding distribution as shown in **Figure 1's Line 3**, which has a detrimental effect on performance.
>
> While other approaches, such as multi-faceted learning strategies, have been suggested, they are often tailored to specific model architectures and can be complex to implement and train. **We contend that the true potential of the IB principle lies not in creating a single, specialized model, but in establishing a new, more effective training paradigm**.
>
> In this paper, we depart from the design of a specific architecture and instead propose a general training paradigm based on the IB theory. **Our approach enhances the global mutual information often ignored by previous models**, enabling them to capture more comprehensive data signals for superior imputation. A key advantage of our paradigm is its flexibility; it is designed as a "plug-in" that can be applied to a wide range of existing TSI models to boost their performance. We have validated this through extensive experiments, demonstrating consistent improvements across various benchmarks.
>
>
>
>
>
> > **W2**: The paper does not clearly report the specific values of $\beta_1$ and $\beta_2$ used in the experiments, nor does it provide practical guidance for selecting them. Given that the sensitivity analysis indicates the method’s performance is strongly affected by these parameters, it would be helpful to include tuning heuristics or recommendations to support practical adoption.
> >
> > **Q2**: How should users determine an appropriate value of $\beta_1$ and $\beta_2$ when applying this method to real-world datasets, especially those with unknown or varying missingness characteristics? Could the authors suggest a principled method, heuristic, or adaptive tuning strategy for selecting $\beta_1$ and $\beta_2$ in practice?
>
> **A**: As illustrated in **Figure 8** of the appendix, our model exhibits consistent sensitivity to hyperparameters across four datasets (ETTh1, ETTh2, ETTm1, ETTm2). The performance trends are remarkably similar, uniformly achieving optimal results when the hyperparameters are set to **0.5, 1e-6, and 0.5**, respectively. In accordance with this finding, we adopted this single, robust set of hyperparameters for all experiments conducted on the nine datasets presented in the main paper. We appreciate the constructive feedback and will incorporate these revisions into the final version of our manuscript.
>
>
>
> > **W3**: The paper claims that high missing rates lead to memorization, where models achieve low training loss but fail to generalize due to poor semantic representations. However, Figure 1 does not clearly demonstrate increased overfitting under high missingness; in fact, the training dynamics appear similar across missingness levels. This raises concerns about the strength of the empirical evidence supporting the central hypothesis.
>
> **A**: **Figure 1** presents the training and testing MAE for all distributions. Taking TimesNet trained for 30 epochs as a representative example (**Line 1**), we observe a clear trend. As the missing rate increases, the training MAE rises slowly from 0.07 to 0.12—an increase of only 0.05. In stark contrast, the test MAE escalates dramatically from 0.36 to 0.81, a jump of 0.45. This growing disparity between training and test performance becomes increasingly pronounced.
>
> Concurrently, we observe that the learned embedding distribution progressively distorts from the original data distribution. The degree of this distortion is exacerbated as the missing rate climbs.
>
> These two phenomena—**the widening performance gap** and **the increasing distributional dissimilarity**—provide strong evidence of a deepening overfitting to noise and a failure to capture high-level semantic information. This empirical result substantiates our core hypothesis.

---

### Note · Authors · 2025-08-12

We are grateful to the reviewers for their insightful comments. They affirmed the strengths of our work as follows:

- **Problem Diagnosis:** Our paper identifies a critical paradox in Time Series Imputation (TSI), where lower training loss at high missing rates leads to distorted embeddings and poor performance.
- **Novelty & Theory:** Our method effectively extends the Information Bottleneck (IB) principle to a novel and theoretically-grounded training paradigm, which aligns latent distributions to preserve global structure while maintaining local accuracy.
- **Empirical Validation:** Extensive experiments on nine datasets show consistent improvements over baselines, particularly in high-missingness scenarios. Visualizations provide strong empirical support for the method's ability to preserve global information.
- **Significance & Impact:** Our work addresses a significant and practical problem and its model-agnostic "plug-and-play" paradigm offers a robust solution, with potential for high impact in fields like healthcare and finance.

> **To address the major concern most reviewers share regarding our choice of the IB and its effectiveness.**

Our choice of the IB directly stems from the core problem we diagnose: current TSI methods, optimized by MAE/MSE, fail at high missing rates by overfitting to noise and losing sight of the data's overall distribution. This results in the paradox where better training loss yields worse test performance. Therefore, a critical problem is raised: **how to enable TSI models to capture both global and local information from incomplete data without overfitting to noise.** The IB principle is uniquely suited to solve this, as its fundamental goal is to capture essential information while discarding noise.

Furthermore, different from current IB-based methods, **we contend that the true potential of the IB principle lies not in creating a single, specialized model, but in establishing a new, more effective training paradigm**. Our "plug-in" approach is designed to enhance the crucial global information that other models miss, enabling existing models to achieve superior imputation.

To validate effectiveness, we extended our paradigm to TCN and DLinear and expanded our baseline comparisons as detailed in the **Tables in Reviewer cTKT's W1 and 3qMw's Q6**, showing consistent improvements across all benchmarks.

*We have addressed the concerns and polished our paper in next version to make our work better understandable.*

---

### Decision · Program_Chairs · 2025-09-17

**Decision:**

Accept (poster)

**Comment:**

This paper addresses a key challenge in time series imputation: models often attain low training loss under high missing rates but experience significant drops in imputation quality and distorted latent representations at inference. To mitigate this issue, the authors propose Glocal Information Bottleneck (Glocal-IB), a model-agnostic extension of the Information Bottleneck framework. Experiments across diverse datasets show that Glocal-IB consistently improves performance.

In the rebuttal, the authors have carefully addressed most of the concerns raised by the reviewers. Three reviewers gave positive scores on the paper, and the only negative reviewer did not participate in the author-reviewer discussion phase.

I have read all the comments and responses from the reviewers and authors, and believe the strength of the paper is sufficient for acceptance.